# Markovian Sliced Wasserstein Distances: Beyond Independent Projections

**Khai Nguyen**
Department of Statistics and Data Sciences
The University of Texas at Austin
Austin, TX 78712
khainb@utexas.edu

**Tongzheng Ren**
Department of Computer Science
The University of Texas at Austin
Austin, TX 78712
tongzheng@utexas.edu

**Nhat Ho**
Department of Statistics and Data Sciences
The University of Texas at Austin
Austin, TX 78712
minhnhat@utexas.edu

## Abstract

Sliced Wasserstein (SW) distance suffers from redundant projections due to independent uniform random projecting directions. To partially overcome the issue, max K sliced Wasserstein (Max-K-SW) distance ($K \geq 1$), seeks the best discriminative orthogonal projecting directions. Despite being able to reduce the number of projections, the metricity of the Max-K-SW cannot be guaranteed in practice due to the non-optimality of the optimization. Moreover, the orthogonality constraint is also computationally expensive and might not be effective. To address the problem, we introduce a new family of SW distances, named *Markovian sliced Wasserstein* (MSW) distance, which imposes a first-order Markov structure on projecting directions. We discuss various members of the MSW by specifying the Markov structure including the prior distribution, the transition distribution, and the burning and thinning technique. Moreover, we investigate the theoretical properties of MSW including topological properties (metricity, weak convergence, and connection to other distances), statistical properties (sample complexity, and Monte Carlo estimation error), and computational properties (computational complexity and memory complexity). Finally, we compare MSW distances with previous SW variants in various applications such as gradient flows, color transfer, and deep generative modeling to demonstrate the favorable performance of the MSW[1].

## 1 Introduction

Sliced Wasserstein (SW) [7] distance has been well-known as a great alternative statistical distance for Wasserstein distance [57, 49]. In short, SW takes the average of Wasserstein distances between corresponding pairs of one-dimensional projected measures as the distance between the two original measures. Hence, the SW has a low computational complexity compared to the conventional Wasserstein distance due to the closed-form solution of optimal transport in one dimension. When the probability measures have at most $n$ supports, the computational complexity of the SW is only $\mathcal{O}(n \log n)$. This complexity is much lower than the computational complexity $\mathcal{O}(n^3 \log n)$ of Wasserstein distance and the complexity $\mathcal{O}(n^2)$ [1, 34, 35, 33] of entropic Wasserstein [11] (Sinkhorn

---

[1]Code for this paper is published at https://github.com/UT-Austin-Data-Science-Group/MSW.

37th Conference on Neural Information Processing Systems (NeurIPS 2023).

divergence). Moreover, the memory complexity of the SW is $\mathcal{O}(n)$ which is lower than $\mathcal{O}(n^2)$ of the Wasserstein (Sinkhorn) distance. The reason is that SW does not need to store the cost matrix between supports which cost $\mathcal{O}(n^2)$. An additional appealing property of the SW is that it does not suffer from the curse of dimensionality, namely, its sample complexity is $\mathcal{O}(n^{-1/2})$ [40, 46] compared to $\mathcal{O}(n^{-1/d})$ [19] of the Wasserstein distance ($d$ is the number of dimensions).

Due to the scalability, the SW has been applied to almost all applications where the Wasserstein distance is used. For example, we refer to some applications of the SW which are generative modeling [60, 15, 27, 42], domain adaptation [30], clustering [28], approximate Bayesian computation [39], gradient flows [37, 5], and variational inference [61]. Moreover, there are many attempts to improve the SW. The generalized sliced Wasserstein (GSW) distance that uses non-linear projection is proposed in [26]. Distributional sliced Wasserstein distance is proposed in [44, 45] by replacing the uniform distribution on the projecting directions in SW with an estimated distribution that puts high probabilities for discriminative directions. Spherical sliced Wasserstein which is defined between distributions that have their supports on the hyper-sphere is introduced in [4]. A sliced Wasserstein variant between probability measures over images with convolution is defined in [43].

Despite having a lot of improvements, one common property in previous variants of the SW is that they use independent projecting directions that are sampled from a distribution over a space of projecting direction e.g., the unit-hypersphere. Those projecting directions are further utilized to project two interested measures to corresponding pairs of one-dimensional measures. Due to the independence, practitioners have reported that many projections do not have the power to discriminative between two input probability measures [26, 15]. Moreover, having a lot of projections leads to redundancy and losing computation for uninformative pairs of projected measures. This problem is known as the projection complexity limitation of the SW.

To partially address the issue, the max sliced Wasserstein (Max-SW) distance is introduced in [14]. Max-SW seeks the best projecting direction that can maximize the projected Wasserstein distance. Since the Max-SW contains a constraint optimization problem, the projected subgradient ascent algorithm is performed. Since the algorithm only guarantees to obtain local maximum [46], the performance of empirical estimation Max-SW is not stable in practice [42] since the metricity of Max-SW can be only obtained at the global optimum. Another approach is to force the orthogonality between projecting directions. In particular, K-sliced Wasserstein [50] (K-SW) uses $K > 1$ orthogonal projecting directions. Moreover, to generalize the Max-SW and the K-SW, max-K sliced Wasserstein (Max-K-SW) distance ($K > 1$) appears in [12] to find the best $K$ projecting directions that are orthogonal to each other via the projected sub-gradient ascent algorithm. Nevertheless, the orthogonality constraint is computationally expensive and might not be good in terms of reflecting discrepancy between general measures. Moreover, Max-K-SW also suffers from the non-optimality problem which leads to losing the metricity property in practice.

To avoid the independency and to satisfy the requirement of creating informative projecting directions efficiently, we propose to impose a *sequential structure* on projecting directions. Namely, we choose a new projecting direction based on the previously chosen directions. For having more efficiency in computation, we consider *first-order Markovian* structure in the paper which means that a projecting direction can be sampled by using only the previous direction. For the first projecting direction, it can follow any types of distributions on the unit-hypersphere that were used in the literature e.g., uniform distribution [7] and von Mises-Fisher distribution [23, 45] to guarantee the metricity. For the transition distribution on the second projecting direction and later, we propose two types of family which are *orthogonal-based* transition and *input-aware* transition. For the orthogonal-based transition, we choose the projecting direction uniformly on the unit hypersphere such that it is orthogonal to the previous direction. In contrast to the previous transition which does not use the information from the two input measures, the input-aware transition uses the sub-gradient with respect to the previous projecting direction of the corresponding projected Wasserstein distance between the two measures to design the transition. In particular, the projected sub-gradient update is used to create the new projecting direction. Moreover, we further improve the computational time and computational memory by introducing the burning and thinning technique to reduce the number of random projecting directions.

**Contribution.** In summary, our contributions are two-fold:

1. We propose a novel family of distances on the space of probability measures, named *Markovian sliced Wasserstein* (MSW) distances. MSW considers a first-order Markovian

structure on random projecting directions. Moreover, we derive three variants of MSW that use two different types of conditional transition distributions: *orthogonal-based* and *input-aware*. We investigate the theoretical properties of MSW including topological properties (metricity, weak convergence, and connection to other distances), statistical properties (sample complexity, and Monte Carlo estimation error), and computational properties (computational complexity and memory complexity). Moreover, we introduce a burning and thinning approach to further reduce computational and memory complexity, and we discuss the properties of the resulting distances.

2. We conduct experiments to compare MSW with SW, Max-SW, K-SW, and Max-K-SW in various applications, namely, gradient flows, color transfer, and deep generative models on standard image datasets: CIFAR10 and CelebA. We show that the input-aware MSW can yield better qualitative and quantitative performance while consuming less computation than previous distances in gradient flows and color transfer, and comparable computation in deep generative modeling. Finally, we investigate the role of hyper-parameters of distances e.g., the number of projections, the number of time-steps, and so on, in applications.

**Organization.** We first provide background for Wasserstein distance, sliced Wasserstein distance, and max sliced Wasserstein distance in Section 2. In Section 3, we propose Markovian sliced Wasserstein distances and derive their theoretical properties. Section 4 contains the comparison of MSW to previous SW variants in gradient flows, color transfer, and deep generative modeling. We then conclude the paper in Section 5. Finally, we defer the proofs of key results in the paper and supplementary materials to Appendices.

**Notation.** For $p \geq 1$, $\mathcal{P}_p(\mathbb{R}^d)$ is the set of all probability measures on $\mathbb{R}^d$ that have finite $p$-moments. For any $d \geq 2$, we denote $\mathcal{U}(\mathbb{S}^{d-1})$ is the uniform measure over the unit hyper-sphere $\mathbb{S}^{d-1} := \{\theta \in \mathbb{R}^d \mid ||\theta||_2^2 = 1\}$. For any two sequences $a_n$ and $b_n$, the notation $a_n = \mathcal{O}(b_n)$ means that $a_n \leq C b_n$ for all $n \geq 1$, where $C$ is some universal constant. We denote $\theta \sharp \mu$ is the push-forward measures of $\mu$ through the function $f : \mathbb{R}^d \to \mathbb{R}$ that is $f(x) = \theta^\top x$.

## 2 Background

We start with reviewing the background on Wasserstein distance, sliced Wasserstein distances, their computation techniques, and their limitations.

**Wasserstein distance.** Given two probability measures $\mu \in \mathcal{P}_p(\mathbb{R}^d)$ and $\nu \in \mathcal{P}_p(\mathbb{R}^d)$, the Wasserstein distance [57, 48] between $\mu$ and $\nu$ is :

$$W_p^p(\mu, \nu) = \inf_{\pi \in \Pi(\mu,\nu)} \int_{\mathbb{R}^d \times \mathbb{R}^d} ||x - y||_p^p d\pi(x, y), \tag{1}$$

where $\Pi(\mu, \nu)$ is set of all couplings that have marginals are $\mu$ and $\nu$ respectively. The computational complexity and memory complexity of Wasserstein distance are $\mathcal{O}(n^3 \log n)$ and $\mathcal{O}(n^2)$ in turn when $\mu$ and $\nu$ have at most $n$ supports. When $d = 1$, the Wasserstein distance can be computed with a closed form: $W_p^p(\mu, \nu) = \int_0^1 |F_\mu^{-1}(z) - F_\nu^{-1}(z)|^p dz$, where $F_\mu$ and $F_\nu$ are the cumulative distribution function (CDF) of $\mu$ and $\nu$ respectively.

**Sliced Wasserstein distance.** By randomly projecting two interested high-dimensional measures to corresponding pairs of one-dimensional measures, sliced Wasserstein (SW) distance can exploit the closed-form benefit of Wasserstein distance in one dimension. The definition of sliced Wasserstein distance [7] between two probability measures $\mu \in \mathcal{P}_p(\mathbb{R}^d)$ and $\nu \in \mathcal{P}_p(\mathbb{R}^d)$ is:

$$SW_p^p(\mu, \nu) = \mathbb{E}_{\theta \sim \mathcal{U}(\mathbb{S}^{d-1})} W_p^p(\theta \sharp \mu, \theta \sharp \nu). \tag{2}$$

Monte Carlo samples are often used to approximate the intractable expectation unbiasedly:

$$\widehat{SW}_p^p(\mu, \nu) = \frac{1}{L} \sum_{l=1}^{L} W_p^p(\theta_l \sharp \mu, \theta_l \sharp \nu), \tag{3}$$

where $\theta_1, \ldots, \theta_L$ are drawn randomly from $\mathcal{U}(\mathbb{S}^{d-1})$. When $\mu$ and $\nu$ are discrete measures that have at most $n$ supports in $d$ dimension, the computational complexity of SW is $\mathcal{O}(Ln \log_2 n + Ldn)$ and the memory complexity for storing the projecting directions and the projected supports is $\mathcal{O}(L(d+n))$.

Here, $Ln\log_2 n$ is for sorting $L$ sets of projected supports and $Ld$ is for projecting supports to $L$ sets of scalars.

**Max sliced Wasserstein distance.** To select the best discriminative projecting direction, the max sliced Wasserstein (Max-SW) distance [14] between $\mu \in \mathcal{P}_p(\mathbb{R}^d)$ and $\nu \in \mathcal{P}_p(\mathbb{R}^d)$ is introduced as follows:

$$\text{Max-SW}_p(\mu,\nu) = \max_{\theta\in\mathbb{S}^{d-1}} W_p(\theta\sharp\mu, \theta\sharp\nu). \tag{4}$$

Computing Max-SW requires solving the constrained optimization problem. In practice, the projected sub-gradient ascent algorithm with $T > 1$ iterations is often used to obtain a surrogate projecting direction $\hat{\theta}_T$ for the global optimum. Hence, the empirical Max-SW distance is $\widehat{\text{Max-SW}}_p(\mu,\nu) = W_p(\hat{\theta}_T\sharp\mu, \hat{\theta}_T\sharp\nu)$. The detail of the projected sub-gradient ascent algorithm is given in Algorithm 1 in Appendix A.1. The computational complexity of Max-SW is $\mathcal{O}(Tn\log_2 n + Tdn)$ and the memory complexity of Max-SW is $\mathcal{O}(d + n)$. It is worth noting that the projected sub-gradient ascent can only yield local maximum [46]. Therefore, the empirical Max-SW might not be distance even when $T \to \infty$ since the metricity of Max-SW can be only obtained at the global maximum.

**K-sliced Wasserstein distance.** The authors in [50] propose to estimate the sliced Wasserstein distance based on orthogonal projecting directions. We refer to the distance as K-sliced Wasserstein distance (K-SW). The definition of K-SW between two probability measures $\mu \in \mathcal{P}_p(\mathbb{R}^d)$ and $\nu \in \mathcal{P}_p(\mathbb{R}^d)$ is:

$$\text{K-SW}_p^p(\mu,\nu) = \mathbb{E}\left[\frac{1}{K}\sum_{i=1}^{K} \text{W}_p^p(\theta_i\sharp\mu, \theta_i\sharp\nu)\right], \tag{5}$$

where the expectation is with respect to $(\theta_1,\ldots,\theta_K) \sim \mathcal{U}(\mathbb{V}_k(\mathbb{R}^d))$ with $\mathbb{V}_K(\mathbb{R}^d) = \{(\theta_1,\ldots,\theta_K) \in \mathbb{S}^{d-1} | \langle\theta_i,\theta_j\rangle = 0 \; \forall i,j \leq K\}$ is the Stiefel manifold. The expectation can be approximated with Monte Carlo samples $(\theta_{1l},\ldots,\theta_{Kl})_{l=1}^L$ from $\mathcal{U}(\mathbb{V}_K(\mathbb{R}^d))$. In the original paper, $L$ is set to 1. To sample from the uniform distribution over the Stiefel manifold $\mathcal{U}(\mathbb{V}_k(\mathbb{R}^d))$, it requires using the Gram-Schmidt orthogonality process which has the computational complexity $\mathcal{O}(K^2 d)$ (quadratic in $K$). Therefore, the total computational complexity of K-SW is $\mathcal{O}(LKn\log_2 n + LKdn + LK^2 d)$ and the memory complexity of K-SW is $\mathcal{O}(LK(d+n))$. More detail related to K-SW including Gram-Smith process and sampling uniformly from the Stiefel manifold is given in Appendix A.1.

**Max K sliced Wasserstein distance.** To generalize both Max-SW and K-SW, Max K sliced Wasserstein is introduced in [12]. Its definition between $\mu \in \mathcal{P}_p(\mathbb{R}^d)$ and $\nu \in \mathcal{P}_p(\mathbb{R}^d)$ is:

$$\text{Max-K-SW}_p^p(\mu,\nu) = \max_{(\theta_1,\ldots,\theta_K)\in\mathbb{V}_K(\mathbb{R}^d)} \left[\frac{1}{K}\sum_{i=1}^{K} \text{W}_p^p(\theta_i\sharp\mu, \theta_i\sharp\nu)\right]. \tag{6}$$

Similar to Max-SW, a projected sub-gradient ascent algorithm with $T > 1$ iterations is used to approximate Max-K-SW. We refer the reader to Algorithm 4 in Appendix A.1 for greater detail. Since the projecting operator to the Stiefel manifold is the Gram-Smith process, the computational complexity of Max-K-SW is $\mathcal{O}(TKn\log_2 n + TKdn + TK^2 d)$. The memory complexity of Max-K-SW is $\mathcal{O}(K(d+n))$. Similar to Max-SW, the metricity of Max-K-SW is only obtained at the global optimum, hence, the empirical estimation might not be stable. Moreover, the orthogonality constraint is also computationally expensive i.e., quadratic in terms of the number of orthogonal projections $K$.

## 3 Markovian Sliced Wasserstein distances

We first define *Markovian sliced Wasserstein* (MSW) distance and discuss its theoretical properties including topological properties, statistical properties, and computational properties in Section 3.1. In Section 3.2, we discuss some choices in designing the Markov chain including the prior distribution and the transition distribution. Finally, we discuss the burning and thinning variant of MSW which can reduce the computational and memory complexity in Section 3.3.

## 3.1 Definitions, Topological, Statistical, and Computational Properties

We first start with a general definition of Markovian sliced Wasserstein distance in Definition 1.

**Definition 1.** *For any $p \geq 1$, $T \geq 1$, and dimension $d \geq 1$, the* Markovian sliced Wasserstein *of order $p$ between two probability measures $\mu \in \mathcal{P}_p(\mathbb{R}^d)$ and $\nu \in \mathcal{P}_p(\mathbb{R}^d)$ is:*

$$MSW_{p,T}^p(\mu,\nu) = \mathbb{E}_{(\theta_{1:T}) \sim \sigma(\theta_{1:T})} \left[ \frac{1}{T} \sum_{t=1}^{T} W_p^p(\theta_t \sharp \mu, \theta_t \sharp \nu) \right], \tag{7}$$

*where $T$ is the number of time steps, the expectation is under the projecting distribution $\theta_{1:T} \sim \sigma(\theta_{1:T})$ with $\sigma(\theta_{1:T}) = \sigma(\theta_1, \ldots, \theta_T) = \sigma_1(\theta_1) \prod_{l=2}^{T} \sigma_t(\theta_t|\theta_{t-1})$, and $\sigma_1(\theta_1), \sigma_t(\theta_t|\theta_{t-1}) \in \mathcal{P}(\mathbb{S}^{d-1})$ for all $t = 1, \ldots, T$.*

The first projecting direction $\theta_1$ follows the distribution $\sigma_1(\theta_1)$ with $\sigma_1(\theta_1)$ to be any distributions on the unit hyper-sphere, e.g., the uniform distribution, a von Mises-Fisher distribution, and so on. By designing the transition distribution $\sigma_l(\theta_l|\theta_{l-1})$, we can obtain various variants of MSW. It is worth noting that the MSW can be rewritten as the average of $T$ expectation of one-dimensional Wasserstein distance, $MSW_{p,T}^p(\mu,\nu) = \frac{1}{T} \sum_{t=1}^{T} \mathbb{E}_{\theta_t \sim \sigma_t(\theta_t)}[W_p^p(\theta_t \sharp \mu, \theta_t \sharp \nu)]$, however, $\sigma_t(\theta_t) = \int \prod_{i=1}^{t} \sigma_1(\theta_1) \prod_{l=2}^{t} \sigma_{t'}(\theta_{t'}|\theta_{t'-1}) d\theta_1 \ldots d\theta_{t-1}$ are not the same for $t = 1, \ldots, T$. Moreover, sampling directly from $\sigma_t(\theta_t)$ is intractable, hence, using the definition of the MSW in Definition 1 is more reasonable in terms of approximating the expectation using Monte Carlo samples.

**Monte Carlo estimation.** Similar to SW, we also need to use Monte Carlo samples to approximate the expectation in Definition 1. We first samples $\theta_{11}, \ldots, \theta_{L1} \sim \sigma_1(\theta_1)$ for $L \geq 1$, then we samples $\theta_{lt} \sim \sigma_t(\theta_t|\theta_{lt-1})$ for $t = 1, \ldots, T$ and $l = 1, \ldots, L$. After that, we can form an unbiased empirical estimation of MSW as follows: $\widehat{MSW}_{p,T}^p(\mu,\nu) = \frac{1}{LT} \sum_{l=1}^{L} \sum_{t=1}^{T} W_p^p(\theta_{lt} \sharp \mu, \theta_{lt} \sharp \nu)$.

Before going to the specific design of those distributions, we first discuss the empirical estimation of MSW, and investigate its theoretical properties including topological properties, statistical properties, and computational properties.

**Topological Properties.** We first state the following assumption: **A1.** *In MSW, the prior distribution $\sigma_1(\theta_1)$ is supported on all the unit-hypersphere or there exists a transition distribution $\sigma_t(\theta_t|\theta_{t-1})$ being supported on all the unit-hypersphere.* The assumption **A1** is easy to satisfy and it holds for all later choices of the prior distribution and transition distribution. We now consider the metricity properties of the Markovian sliced Wasserstein distance.

**Theorem 1** (Metricity). *For any $p \geq 1$, $T \geq 1$, and dimension $d \geq 1$, if **A1** holds, Markovian sliced Wasserstein $MSW_{p,T}(\cdot,\cdot)$ is a valid metric on the space of probability measures $\mathcal{P}_p(\mathbb{R}^d)$, namely, it satisfies the (i) non-negativity, (ii) symmetry, (iii) triangle inequality, and (iv) identity.*

The proof of Theorem 1 is in Appendix B.1. Next, we show that the convergence in MSW implies the weak convergence of probability measures and the reverse also holds.

**Theorem 2** (Weak Convergence). *For any $p \geq 1$, $T \geq 1$, and dimension $d \geq 1$, if **A1** holds, the convergence of probability measures in $\mathcal{P}_p(\mathbb{R}^d)$ under the Markovian sliced Wasserstein distance $MSW_{p,T}(\cdot,\cdot)$ implies weak convergence of probability measures and vice versa.*

Theorem 2 means that for any sequence of probability measures $(\mu_k)_{k \in \mathbb{N}}$ and $\mu$ in $\mathcal{P}_p(\mathbb{R}^d)$, $\lim_{k \to +\infty} MSW_{p,T}(\mu_k, \mu) = 0$ if and only if for any continuous and bounded function $f : \mathbb{R}^d \to \mathbb{R}$, $\lim_{k \to +\infty} \int f \, d\mu_k = \int f \, d\mu$. The proof of Theorem 2 is in Appendix B.2. Next, we discuss the connection of MSW to previous sliced Wasserstein variants.

**Proposition 1.** *For any $p \geq 1$ and dimension $d \geq 1$,*

*(i) For any $T \geq 1$ and $\mu, \nu \in \mathcal{P}_p(\mathbb{R}^d)$, $MSW_{p,T}(\mu,\nu) \leq Max\text{-}SW_p(\mu,\nu) \leq W_p(\mu,\nu)$.*

*(ii) If $T = 1$ and the prior $\sigma_1(\theta_1) := \mathcal{U}(\mathbb{S}^{d-1})$, $MSW_{p,T}(\mu,\nu) = SW_p(\mu,\nu)$.*

The proof of Proposition 1 is in Appendix B.3.

**Statistical Properties.** We investigate the sample complexity (empirical estimation rate) of the MSW.

**Proposition 2** (Sample Complexity). *Let $X_1, X_2, \ldots, X_n$ be i.i.d. samples from the probability measure $\mu$ being supported on compact set of $\mathbb{R}^d$. We denote the empirical measure $\mu_n = \frac{1}{n}\sum_{i=1}^n \delta_{X_i}$. Then, for any $p \geq 1$ and $T \geq 1$, there exists a universal constant $C > 0$ such that $\mathbb{E}[MSW_{p,T}(\mu_n, \mu)] \leq C\sqrt{(d+1)\log n / n}$, where the outer expectation is taken with respect to the data $X_1, X_2, \ldots, X_n$.*

The proof of Proposition 2 is in Appendix B.4. The proposition suggests that MSW does not suffer from the curse of dimensionality. Next, we investigate the MSW's Monte Carlo approximation error.

**Proposition 3** (Monte Carlo error). *For any $p \geq 1$, $T \geq 1$, dimension $d \geq 1$, and $\mu, \nu \in \mathcal{P}_p(\mathbb{R}^d)$, we have: $\mathbb{E}|\widehat{MSW}_{p,T}^p(\mu, \nu) - MSW_{p,T}^p(\mu, \nu)| \leq \frac{1}{T\sqrt{L}} Var\left[\sum_{t=1}^T W_p^p(\theta_t \sharp \mu, \theta_t \sharp \nu)\right]^{\frac{1}{2}}$, where the variance is with respect to $\sigma(\theta_1, \ldots, \theta_T)$.*

The proof of Proposition 3 is in Appendix B.5. From the above proposition, we know that increasing the number of projections $L$ reduces the approximation error.

**Computational Properties.** When $\mu$ and $\nu$ are two discrete probability measures in $\mathcal{P}_p(\mathbb{R}^d)$ that have at most $n$ supports, the computational complexity for the Monte Carlo approximation of MSW is $\mathcal{O}(TLn \log_2 n + TLdn)$ where $\mathcal{O}(TLn \log n)$ is for computation of $TL$ one-dimensional Wasserstein distances and $\mathcal{O}(TLdn)$ is the projecting complexity for $TL$ projections from $d$ dimension to 1 dimension. The memory complexity of MSW is $\mathcal{O}(TL(d+n))$ for storing the projecting directions and the projections.

## 3.2 Specific Choices of Projecting Distributions

Designing the projecting distribution $\sigma(\theta_1, \ldots, \theta_T)$ is the central task in using MSW since it controls the projecting behavior. For each choice of the $\sigma(\theta_1, \ldots, \theta_T)$, we obtain a variant of MSW. Since we impose the first order Markov structure $\sigma(\theta_1, \ldots, \theta_T) = \sigma_1(\theta_1) \prod_{t=2}^T \sigma_t(\theta_t|\theta_{t-1})$, there are two types of distributions that we need to choose: the *prior distribution* $\sigma_1(\theta_1)$ and the *transition distribution* $\sigma_t(\theta_t|\theta_{t-1})$ for all $t = 2, \ldots, T$.

**Prior distribution.** The most simple choice of $\sigma_1(\theta_1)$ when we know nothing about probability measures that we want to compare is the uniform distribution over the unit hypersphere $\mathcal{U}(\mathbb{S}^{d-1})$. Moreover, the metricity of MSW is guaranteed regardless of the transition distribution with this choice. Therefore, the uniform distribution is the choice that we use in our experiments in the paper. It is worth noting that we could also use a distribution that is estimated from two interested probability measures [44]; however, this approach costs more computation.

**Transition distribution.** We discuss some specific choices of the transition distributions $\sigma_t(\theta_t|\theta_{t-1})$. Detailed algorithms for computing MSW with specific transitions are given in Appendix A.3.

*Orthogonal-based transition.* Motivated by the orthogonality constraint in Max-K-SW and K-SW, we can design a transition distribution that gives us an orthogonal projecting direction to the previous one. In particular, given a previous projecting direction $\theta_{t-1}$, we want to have $\theta_t$ such that $\langle \theta_t, \theta_{t-1} \rangle = 0$, namely, we want to sample from the subsphere $\mathcal{S}_{\theta_{t-1}}^{d-1} := \{\theta_t \in \mathbb{S}^{d-1} | \langle \theta_t, \theta_{t-1} \rangle = 0\}$. To the best of our knowledge, there is no explicit form of distribution (known pdf) that is defined on that set. However, we can still sample from the uniform distribution over that set: $\mathcal{U}(\mathcal{S}_{\theta_{t-1}}^{d-1})$ since that distribution can be constructed by pushing the uniform distribution over the whole unit hypersphere $\mathcal{U}(\mathbb{S}^{d-1})$ through the projection operator: $\text{Proj}_{\theta_{t-1}}(\theta_t) = \text{Proj}_{\mathbb{S}^{d-1}}\left(\theta_t - \frac{\langle \theta_{t-1}, \theta_t \rangle}{\langle \theta_{t-1}, \theta_{t-1} \rangle} \theta_{t-1}\right)$ where $\text{Proj}_{\mathbb{S}^{d-1}}(\theta) = \frac{\theta}{||\theta||_2}$ is the normalizing operator. In a greater detail, we first sample $\theta_t' \sim \mathcal{U}(\mathbb{S}^{d-1})$ and then set $\theta_t = \text{Proj}_{\theta_{t-1}}(\theta_t')$. Therefore, we have $\sigma_t(\theta_t|\theta_{t-1}) = \mathcal{U}(\mathcal{S}_{t-1}^{d-1}) = \text{Proj}_{\theta_{t-1}} \sharp \mathcal{U}(\mathbb{S}^{d-1})$.

*Input-aware transition.* The above two transition distributions do not take into account the information of the two probability measures $\mu$ and $\nu$ that we want to compare. Hence, it could be inefficient to explore good projecting directions by comparing $\mu$ and $\nu$. Motivated by the projected sub-gradient ascent [9] update in finding the "max" projecting direction, we could design the transition distribution as follows: $\sigma_t(\theta_t|\theta_{t-1}) = \delta_{f(\theta_{t-1}|\eta, \mu, \nu)}$ where $\delta$ denotes the Dirac Delta function and the transition function $f(\theta_{t-1}|\eta, \mu, \nu) = \text{Proj}_{\mathbb{S}^{d-1}}\left(\theta_{t-1} + \eta \nabla_{\theta_{t-1}} W_p(\theta_{t-1}\sharp\mu, \theta_{t-1}\sharp\nu)\right)$, with $\eta > 0$ is the stepsize hyperparameter. As the current choice is a deterministic transition, it requires the prior

distribution to have supports on all $\mathbb{S}^{d-1}$ to obtain the metricity for MSW. A choice to guarantee the metricity regardless of the prior distribution is the von Mises-Fisher (vMF) distribution [23], namely, $\sigma_t(\theta_t|\theta_{t-1}) = \text{vMF}(\theta_t|\epsilon = f(\theta_{t-1}|\eta, \mu, \mu), \kappa)$. The details about the vMF distribution including its probability density function, its sampling procedure, and its properties are given in Appendix A.2. In summary, the vMF distribution has two parameters: the location parameter $\epsilon \in \mathbb{S}^{d-1}$ which is the mean, and the concentration parameter $\kappa \in \mathbb{R}^+$ which plays the role as the precision. Thank the interpolation properties of the vMF distribution: $\lim_{\kappa \to 0} \text{vMF}(\theta|\epsilon, \kappa) = \mathcal{U}(\mathbb{S}^{d-1})$ and $\lim_{\kappa \to \infty} \text{vMF}(\theta|\epsilon, \kappa) = \delta_\epsilon$, the transition distribution can balance between heading to the "max" projecting direction and exploring the space of directions.

**Stationarity of $\sigma_T(\theta_T)$.** A natural important question arises: what is the distribution of $\sigma_T(\theta_T) = \int \ldots \int \sigma(\theta_1, \ldots, \theta_T) d\theta_1 \ldots d\theta_{T-1}$ when $T \to \infty$? The answer to the above questions depends on the choice of the projection distribution which is discussed in Section 3.2. For the *Orthogonal-based* transitions and the uniform distribution prior, it is unclear whether the stationary distribution exists. For the deterministic *Input-aware* transition and the uniform prior, we have $\lim_{T \to \infty} \sigma_T(\theta_T) = \sum_{a=1}^{A} \alpha_a \delta_{\theta_a^*}$ with $\sum_{a=1}^{A} \alpha_a = 1$ where $\theta_a^*$ ($a = 1, \ldots, A$) are local maximas of the optimization problem $\max_{\theta \in \mathbb{S}^{d-1}} W_p(\theta\sharp\mu, \theta\sharp\nu)$ and some unknown weights $\alpha_a$ that depend on $\mu$ and $\nu$. This property is due to the fact that the projected sub-gradient ascent can guarantee local maxima convergence [46]. For the *Input-aware* vMF transition, it is also unclear if the stationary distribution exists when the parameter $\kappa < \infty$.

## 3.3 Burning and Thinning

In the definition of MSW in Definition 1, we take the expectation on the joint distribution over all timesteps $\sigma(\theta_{1:T})$ which leads to the time and memory complexities to be linear with $T$ in the Monte Carlo approximation. Therefore, we can adapt the practical technique from MCMC methods which is burning and thinning to reduce the number of random variables while still having the dependency.

**Definition 2.** *For any $p \geq 1$, $T \geq 1$, dimension $d \geq 1$, the number of burned steps $M \geq 0$, and the number of thinned steps $N \geq 1$, the* burned thinned Markovian sliced Wasserstein *of order $p$ between two probability measures $\mu \in \mathcal{P}_p(\mathbb{R}^d)$ and $\nu \in \mathcal{P}_p(\mathbb{R}^d)$ is:*

$$MSW_{p,T,N,M}^p(\mu, \nu) = \mathbb{E}\left[\frac{N}{T-M} \sum_{t=1}^{(T-M)/N} W_p^p(\theta_t'\sharp\mu, \theta_t'\sharp\nu)\right], \tag{8}$$

*where the expectation is under the projection distribution $\theta_{1:N(T-M)}' \sim \sigma(\theta_{1:N(T-M)}')$ with $\sigma(\theta_{1:N/(T-M)}')$ being the marginal distribution which is obtained by integrating out random projecting directions at the time step $t$ such that $t \leq M$ or $t\%N \neq 0$ from $\sigma(\theta_1, \ldots, \theta_T) = \sigma_1(\theta_1) \prod_{l=2}^{T} \sigma_t(\theta_t|\theta_{t-1})$ for all $t = 1, \ldots, T$.*

Similar to MSW, the burned-thinned MSW is also a metric on $\mathcal{P}_p(\mathbb{R}^d)$ when there exists a time step $t$ that is not burned, is not thinned, and $\theta_t$ is a random variable that has the supports on all $\mathbb{S}^{d-1}$. We discuss more details about the burned-thinned MSW including its topological and statistical properties in Appendix A.4. The Monte Carlo estimation of the burned-thinned MSW is given in Equation 10 in Appendix A.4. The approximation is the average of the projected Wasserstein distance from $\theta_{tl}$ with $t \geq M$ and $t\%N = 0$. By reducing the number of random projecting directions, the computational complexity of the burned-thinned MSW is improved to $\mathcal{O}(((T-M)Ln\log_2 n + (T-M)Ldn)/N)$ in the orthogonal-based transitions. In the case of the input-aware transition, the computational complexity is still $\mathcal{O}(TLn\log_2 n + TLdn)$ since the transition requires computing the gradient of the projected Wasserstein distance. However, in all cases, the memory complexity is reduced to $\mathcal{O}((T-M)L(d+n)/N)$.

**Burned thinned MSW is the generalization of Max-SW.** the empirical computation of Max-SW [14] with the projected sub-gradient ascent and uniform random initialization can be viewed as a special case of burned thinned MSW with the input-aware transition and with the number of burned samples $M = T - 1$. The difference is that Max-SW uses only one local maximum to compute the distance while the burned thinned MSW uses $L \geq 1$ maximums (might not be unique).

**More discussions.** We refer the reader to Appendix A.5 for other related discussions e.g., "K-SW is autoregressive decomposition of projecting distribution", "sequential generalization of Max-K-SW", and related literature.

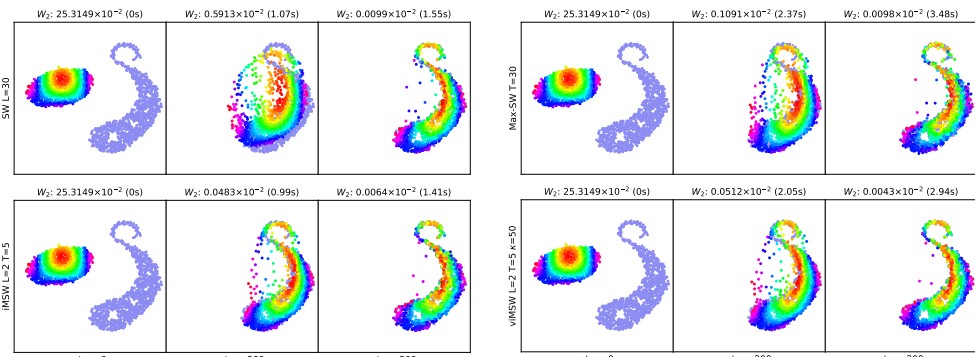

Figure 1: The figures show the gradient flows that are from the empirical distribution over the color points to the empirical distribution over S-shape points. The corresponding Wasserstein-2 distance between the empirical distribution at the current step and the S-shape distribution and the computational time (in seconds) to reach the step is reported at the top of the figure.

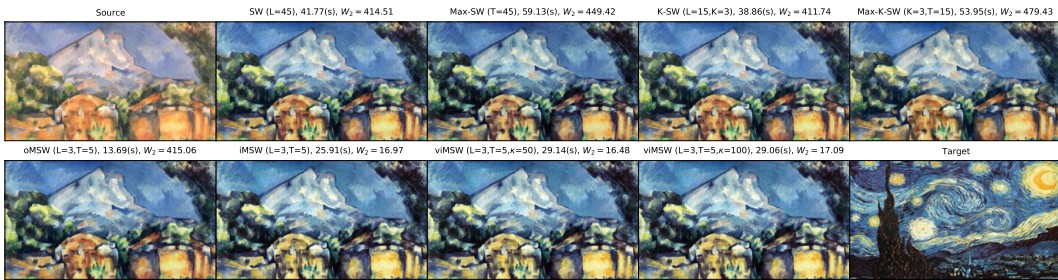

Figure 2: The figures show the source image, the target image, and the transferred images from different distances. The corresponding Wasserstein-2 distance between the empirical distribution over transferred color palates and the empirical distribution over the target color palette and the computational time (in second) are reported at the top of the figure.

## 4 Experiments

In this section, we refer MSW with orthogonal-based transition as oMSW, MSW with input-awared transition as iMSW (using the Dirac distribution) and viMSW (using the vMF distribution). We compare MSW variants to SW, Max-SW, K-SW, and Max-K-SW in standard applications e.g., gradient flows, color transfer, and deep generative models. Moreover, we also investigate the role of hyperparameters, e.g., concentration parameter $\kappa$, the number of projections $L$, the number of time steps $T$, the number of burning steps $M$, and the number of thinning steps $N$ in applications.

### 4.1 Gradient Flows and Color Transfer

**Gradient flows.** We follow the same setting in [17]. The gradient flow models a distribution $\mu(t)$ flowing with time $t$ along the gradient flow of a loss functional $\mu(t) \to \mathcal{D}(\mu(t), \nu)$ that drives it towards a target distribution $\nu$ [53] where $\mathcal{D}$ is a given distance between probability measures. In this setup, we consider $\nu = \frac{1}{n} \sum_{i=1}^{n} \delta_{Y_i}$ as a fixed empirical target distribution and the model distribution $\mu(t) = \frac{1}{n} \sum_{i=1}^{n} \delta_{X_i(t)}$. Here, the model distribution is parameterized by a time-varying point cloud $X(t) = (X_i(t))_{i=1}^{n} \in (\mathbb{R}^d)^n$. Starting from an initial condition at time $t = 0$, we integrate the ordinary differential equation $\dot{X}(t) = -n\nabla_{X(t)} \left[ \mathcal{D} \left( \frac{1}{n} \sum_{i=1}^{n} \delta_{X_i(t)}, \nu \right) \right]$ for each iteration. In the experiments, we utilize the Euler scheme with 300 timesteps and the step size is $10^{-3}$ to move the empirical distribution over colorful 1000 points $\mu(0)$ to the distribution over S-shape 1000 points ($\nu$) (see Figure 1). For Max-SW, Max-K-SW, iMSW, and viMSW, we use the learning rate parameter for projecting directions $\eta = 0.1$. We report the Wasserstein-2 distances between the empirical distribution $\mu(t)$ and the target empirical distribution $\nu$, and the computational time in Table 1. We also give the visualization of some obtained flows in Figure 1. We refer the reader to Figure 4 in Appendix C.1 for the full visualization of all flows and detailed algorithms. We observe that iMSW

Table 1: Summary of Wasserstein-2 distances, computational time in second (s) of different gradient flows.

| Distances | Wasserstein-2 ($\downarrow$) | Time ($\downarrow$) | Distances | Wasserstein-2 ($\downarrow$) | Time ($\downarrow$) |
|---|---|---|---|---|---|
| SW (L=30) | $0.0099 \times 10^{-2}$ | 1.55 | Max-K-SW (K=2,T=15) | $0.0146 \times 10^{-2}$ | 3.35 |
| Max-SW (T=30) | $0.0098 \times 10^{-2}$ | 3.48 | iMSW (L=2,T=5) (ours) | $0.0064 \times 10^{-2}$ | **1.41** |
| K-SW (L=15,K=2) | $0.0098 \times 10^{-2}$ | 1.71 | viMSW (L=2,T=5,$\kappa$=50)(ours) | $\mathbf{0.0043 \times 10^{-2}}$ | 2.94 |

Table 2: Summary of FID and IS scores from three different runs on CIFAR10 (32x32), and CelebA (64x64).

| Method | CIFAR10 (32x32) | | CelebA (64x64) | Method | CIFAR10 (32x32) | | CelebA (64x64) |
|---|---|---|---|---|---|---|---|
| | FID ($\downarrow$) | IS ($\uparrow$) | FID ($\downarrow$) | | FID ($\downarrow$) | IS ($\uparrow$) | FID ($\downarrow$) |
| SW | 14.21$\pm$1.12 | 8.19$\pm$0.07 | 8.93$\pm$0.23 | oMSW (ours) | 14.12$\pm$0.54 | 8.20$\pm$0.05 | 9.68$\pm$0.55 |
| Max-K-SW | 14.38$\pm$0.08 | 8.15$\pm$0.02 | 8.94$\pm$0.35 | iMSW (ours) | 14.12$\pm$0.48 | **8.24$\pm$0.09** | **8.89$\pm$0.23** |
| K-SW | 15.24$\pm$0.02 | 8.15$\pm$0.03 | 9.41$\pm$0.16 | viMSW (ours) | **13.98$\pm$0.59** | 8.12$\pm$0.20 | 8.91$\pm$0.11 |

gives better flows than SW, Max-SW, K-SW, and Max-K-SW. Namely, the empirical distribution $\mu(t)$ ($t = 300$) with iMSW is closer to $\nu$ in terms of Wasserstein distance. More importantly, iMSW consumes less computation than its competitors since it can use a smaller number of projections due to more informative projecting directions. Furthermore, viMSW gives better final results than iMSW, however, the trade-off is doubling the time computation due to the sampling step of vMF distribution. In this case, K-SW is equivalent to our oMSW with T=K=2 since the dimension $d = 2$. We refer the reader to Appendix C.1 for more discussion.

**Studies on hyperparameters.** From Table 3 in Appendix C.1, increasing the number of projections $L$ yields better performance for SW, K-SW, and iMSW. Similarly, increasing the number of timesteps $T$ also helps Max-SW and iMSW better. Moreover, we find that for the same number of total projections e.g., $L = 5, T = 2$ and $T = 2, L = 5$, a larger timestep $T$ might lead to a better result for iMSW. For burning and thinning, we see that they help to reduce the computation while the performance stays comparable or even better if choosing the right value of $M$ and $N$. Also, iMSW the burning steps $M = T - 1$ is still better than Max-SW with $T$ time steps. For the concentration parameter $\kappa$ in viMSW, the performance of viMSW is not monotonic in terms of $\kappa$.

**Color transfer.** We aim to transfer the color palate (RGB) of a source image to the color palette (RGB) target image. Therefore, it is natural to build a gradient flow that starts from the empirical distribution over the color palette of the source image to the empirical distribution over the color palette of the target image. Since the value of color palette is in the set $\{0, \ldots, 255\}^3$, we round the value of the supports of the empirical distribution at the final step of the Euler scheme with 2000 steps and $10^{-3}$ step size. Greater detail can be found in Appendix C.2. For Max-SW, Max-K-SW, iMSW, and viMSW, we use the learning rate parameter for projecting directions $\eta = 0.1$. We show the transferred images, the corresponding Wasserstein-2 distances between the empirical distribution over the transferred color palette and the empirical distribution over the target color palette, and the corresponding computational time in Figure 2. From the figures, iMSW and viMSW give the best transferred images quantitatively and qualitatively. Moreover, oMSW is comparable to SW, Max-SW, K-SW, and are better than Max-K-SW while consuming much less computation. We refer the reader to Figure 5 in Appendix C.2 for the color palette visualization and to Figure 6 for another choice of the source and target images. We also conduct studies on hyperparameters in Appendix C.2 where we observe some similar phenomenons as in gradient flow.

## 4.2 Deep Generative Models

We follow the setup of sliced Wasserstein deep generative models in [15]. The full settings of the framework including neural network architectures, training framework, and hyperparameters are given Appendix C.3. We compare MSW with previous baselines including SW, Max-SW, K-SW, and Max-K-SW on benchmark datasets: CIFAR10 (image size 32x32) [29], and CelebA [36] (image size 64x64). The evaluation metrics are FID score [21] and Inception score (IS) [51] (except on CelebA since IS score poorly captures the perceptual quality of face images [21]). A notable change in computing Max-SW is that we do not use momentum in optimization for max projecting direction like in previous works [26, 42], which leads to a better result.

**Summary of generative performance.** We train generative models with SW ($L \in \{100, 1000, 10000\}$), K-SW ($L \in \{1, 10, 100\}, K = 10$), Max-K-SW ($K = \{1, 10\}, T = \{1, 10, 100, 1000\}, \eta \in \{0.01, 0.1\}$)(Max-K-SW (K=1) is Max-SW), MSW (all variant, $L =$

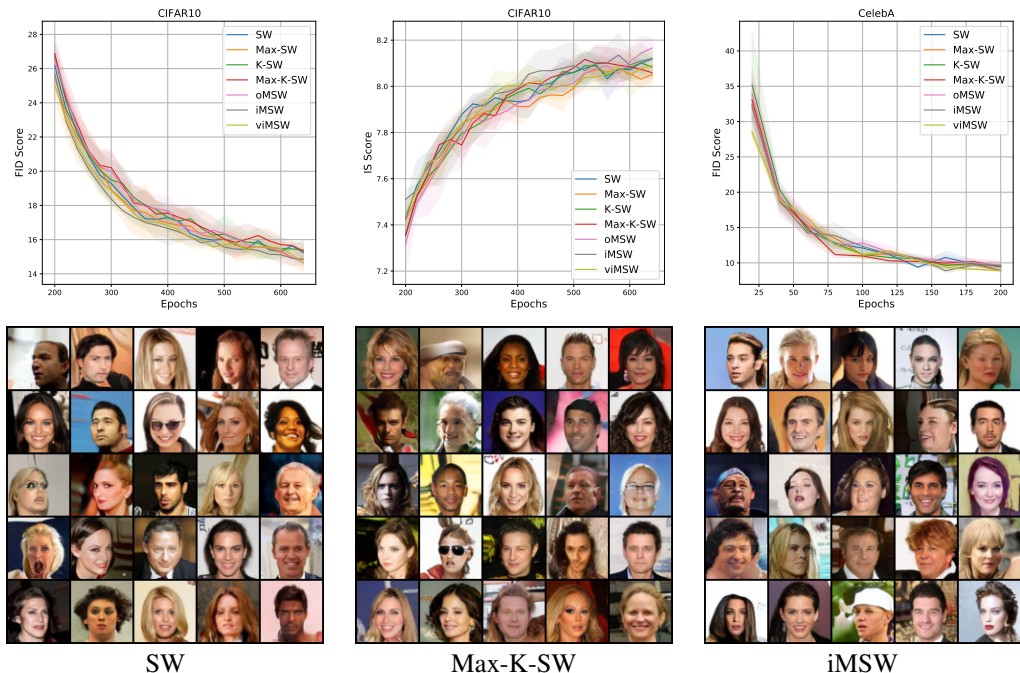

| SW | Max-K-SW | iMSW |

Figure 3: The FID scores and the IS scores over epochs, and some generated images from CelebA.

$\{10, 100\}, T \in \{10, 100\}$), iMSW and viMSW ($\eta \in \{0.01, 0.1\}$), and viMSW and ($\kappa \in \{10, 50\}$). We report the best FID score and the best IS score for each distance in Table 2. In addition, we show how scores change with respect to the training epochs in Figure 3. Overall, we observe that viMSW and iMSW give the best generative performance in terms of the final scores and fast convergence on CIFAR10 and CelebA. The oMSW gives comparable results to baselines. Since most computation in training deep generative models is for updating neural networks, the computational time for distances is almost the same. Furthermore, we show some generated images on CelebA in Figure 3 and all generated images on CIFAR10 and CelebA in Figure 7 and Figure 8 in Appendix C.3. We visually observe that the qualitative results are consistent with the quantitative results in Table 2.

**Studies on hyperparameters.** We conduct experiments to understand the behavior of the burning and thinning technique, and to compare the role of $L$ and $T$ in Table 5 in Appendix C.3. Overall, burning (thinning) sometimes helps to improve the performance of training generative models. There is no clear sign of superiority between burning and thinning. We compare two settings of the same number of total projections (same complexities): $L = 10, T = 100$ and $L = 100, T = 10$. On CIFAR10, the first setting is better while the reverse case happens on CelebA.

## 5  Conclusion

We have introduced the Markovian sliced Wasserstein (MSW), a novel family of sliced Wasserstein (SW) distances, which imposes a first-order Markov structure on projecting directions. We have investigated the theoretical properties of MSW including topological properties, statistical properties, and computational properties. Moreover, we have discussed three types of transition distribution for MSW, namely, orthogonal-based, and input-aware transitions. In addition, we have proposed a burning and thinning technique to improve the computational time and memory of MSW. Finally, we have compared MSW to previous variants of SW in gradient flows, color transfer, and generative modeling to show that MSW distances are both effective and efficient.

## Acknowledgements

NH acknowledges support from the NSF IFML 2019844 and the NSF AI Institute for Foundations of Machine Learning.

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

# Supplement to "Markovian Sliced Wasserstein Distances: Beyond Independent Projections"

In this supplementary material, we present additional materials in Appendix A. In particular, we provide additional background on sliced Wasserstein variants in Appendix A.1, background on von Mises-Fisher distribution in Appendix A.2, algorithms for computing Markovian sliced Wasserstein distances in Appendix A.3, additional information about burned thinned MSW in Appendix A.4, and discussion on related works in Appendix A.5. We then provide skipped proofs in the main paper in Appendix B. Additional experiments are presented in Appendix C.

## A Additional Materials

### A.1 Background on Sliced Wasserstein Variants

We review computational aspects of sliced Wasserstein variants.

**Computation of Max sliced Wasserstein distance.** We demonstrate the empirical estimation of Max-SW via projected sub-gradient ascent algorithm in Algorithm 1. The initialization step for $\hat{\theta}_0$ is rarely discussed in previous works. Normally, $\hat{\theta}_0$ is randomly initialized by drawing from the uniform distribution over the unit-hypersphere. Many previous works [26, 44, 45, 42] use Adam update instead of the standard gradient ascent update for Max-SW. In this work, we find out that using the standard gradient ascent update is more stable and effective.

---

**Algorithm 1** Max sliced Wasserstein distance

**Input.** Probability measures $\mu, \nu$, learning rate $\eta$, the order $p$, and the number of iterations $T$.
Initialize $\hat{\theta}_0$.
**for** $t = 1$ to $T - 1$ **do**
$\quad \hat{\theta}_t = \hat{\theta}_{t-1} + \eta \cdot \nabla_{\hat{\theta}_{t-1}} W_p(\hat{\theta}_{t-1}\sharp\mu, \hat{\theta}_{t-1}\sharp\nu)$
$\quad \hat{\theta}_t = \frac{\hat{\theta}_t}{||\hat{\theta}_t||_2}$
**end for**
**Return.** $W_p(\hat{\theta}_T\sharp\mu, \hat{\theta}_T\sharp\nu)$

---

**K sliced Wasserstein distance.** We first review the Gram–Schmidt process in Algorithm 2. With the Gram–Schmidt process, the sampling from $\mathcal{U}(\mathbb{V}_K(\mathbb{R}^d))$ can be done by sampling $\theta_1, \ldots, \theta_k$ i.i.d from $\mathcal{N}(0, I_d)$ then applying the Gram-Schmidt process on them. Therefore, we present the computation of K sliced Wasserstein distance in Algorithm 3. We would like to recall that the original work of K-SW [50] uses only one set of orthogonal projecting directions. Here, we generalize the original work by using $L$ sets of orthogonal projecting directions.

---

**Algorithm 2** Gram–Schmidt process

**Input.** $K$ vectors $\theta_1, \ldots, \theta_K$
$\theta_1 = \frac{\theta_1}{||\theta_1||_2}$
**for** $k = 2$ to $K$ **do**
$\quad$ **for** $i = 1$ to $k - 1$ **do**
$\quad\quad \theta_k = \theta_k - \frac{\langle\theta_i, \theta_k\rangle}{\langle\theta_i, \theta_i\rangle}\theta_i$
$\quad$ **end for**
$\quad \theta_k = \frac{\theta_k}{||\theta_k||_2}$
**end for**
**Return.** $\theta_1, \ldots, \theta_K$

---

**Max K sliced Wasserstein distance.** We now present the empirical estimation of Max-K-SW via projected sub-gradient ascent algorithm in Algorithm 4. This algorithm is first discussed in the original paper of Max-K-SW [12]. The optimization of Max-K-SW can be solved by using Riemannian optimization since the Stiefel manifold is a Riemannian manifold. However, to the best of our knowledge, Riemannian optimization has not been applied to Max-K-SW.

---

**Algorithm 3** K sliced Wasserstein distance

---

**Input.** Probability measures $\mu, \nu$, the dimension $d$, the order $p$, the number of projections $L$, the number of orthogonal projections $K$.
**for** $l = 1$ to $L$ **do**
     Draw $\theta_{l1}, \ldots, \theta_{lK}$ i.i.d from $\mathcal{N}(0, I_d)$.
     $\theta_{l1}, \ldots, \theta_{lK} =$ Gram–Schmidt$(\theta_{l1}, \ldots, \theta_{lK})$
**end for**
**Return.** $\left( \frac{1}{LK} \sum_{l=1}^{L} \sum_{k=1}^{K} W_p^p(\theta_{lk} \sharp \mu, \theta_{lk} \sharp \nu) \right)^{\frac{1}{p}}$

---

---

**Algorithm 4** Max-K sliced Wasserstein distance

---

**Input.** Probability measures $\mu, \nu$, learning rate $\eta$, the dimension $d$, the order $p$, the number of iterations $T > 1$, and the number of orthogonal projections $K > 1$.
Initialize $\hat{\theta}_{01}, \ldots, \hat{\theta}_{0K}$ to be orthogonal.
**for** $t = 1$ to $T - 1$ **do**
     **for** $k = 1$ to $K$ **do**
         $\hat{\theta}_{tk} = \theta_{tk} + \eta \cdot \nabla_{\hat{\theta}_{t-1k}} W_p(\hat{\theta}_{t-1k} \sharp \mu, \hat{\theta}_{t-1k} \sharp \nu)$
     **end for**
     $\hat{\theta}_{t1}, \ldots, \hat{\theta}_{tK} =$ Gram-Schmidt$(\hat{\theta}_{t1}, \ldots, \hat{\theta}_{tK})$
**end for**
**Return.** $\left( \frac{1}{K} \sum_{k=1}^{K} W_p^p(\hat{\theta}_{Tk} \sharp \mu, \hat{\theta}_{Tk} \sharp \nu) \right)^{\frac{1}{p}}$

---

## A.2 Von Mises-Fisher Distribution

We first start with the definition of von Mises-Fisher (vMF) distribution.

---

**Algorithm 5** Sampling from vMF distribution

---

**Input.** location $\epsilon$, concentration $\kappa$, dimension $d$, unit vector $e_1 = (1, 0, .., 0)$
Draw $v \sim \mathcal{U}(\mathbb{S}^{d-2})$
$b \leftarrow \frac{-2\kappa + \sqrt{4\kappa^2 + (d-1)^2}}{d-1}, a \leftarrow \frac{(d-1) + 2\kappa + \sqrt{4\kappa^2 + (d-1)^2}}{4}, m \leftarrow \frac{4ab}{(1+b)} - (d-1)\log(d-1)$
**repeat**
     Draw $\psi \sim$ Beta $\left( \frac{1}{2}(d-1), \frac{1}{2}(d-1) \right)$
     $\omega \leftarrow h(\psi, \kappa) = \frac{1 - (1+b)\psi}{1 - (1-b)\psi}$
     $t \leftarrow \frac{2ab}{1 - (1-b)\psi}$
     Draw $u \sim \mathcal{U}([0, 1])$
**until** $(d-1)\log(t) - t + m \geq \log(u)$
$h_1 \leftarrow (\omega, \sqrt{1 - \omega^2} v^\top)^\top$
$\epsilon' \leftarrow e_1 - \epsilon$
$u = \frac{\epsilon'}{||\epsilon'||_2}$
$U = I - 2uu^\top$
**Output.** $Uh_1$

---

**Definition 3.** *The von Mises–Fisher distribution (*vMF*)[23] is a probability distribution on the unit hypersphere $\mathbb{S}^{d-1}$ with the density function be:*

$$f(x|\epsilon, \kappa) := C_d(\kappa) \exp(\kappa \epsilon^\top x), \qquad (9)$$

*where $\epsilon \in \mathbb{S}^{d-1}$ is the location vector, $\kappa \geq 0$ is the concentration parameter, and $C_d(\kappa) := \frac{\kappa^{d/2-1}}{(2\pi)^{d/2} I_{d/2-1}(\kappa)}$ is the normalization constant. Here, $I_v$ is the modified Bessel function of the first kind at order $v$ [56].*

The vMF distribution is a continuous distribution, its mass concentrates around the mean $\epsilon$, and its density decrease when $x$ goes away from $\epsilon$. When $\kappa \to 0$, vMF converges in distribution to $\mathcal{U}(\mathbb{S}^{d-1})$, and when $\kappa \to \infty$, vMF converges in distribution to the Dirac distribution centered at $\epsilon$ [55].

**Sampling:** We review the sampling process in Algorithm 5 [13, 45]. The sampling process of vMF distribution is based on the rejection sampling procedure. It is worth noting that the sampling algorithm is doing reparameterization implicitly. However, we only use the algorithm to obtain random samples without estimating stochastic gradients.

## A.3 Algorithms for Computing Markovian Sliced Wasserstein Distances

We first start with the general computation of MSW in Algorithm 6. For the orthogonal-based transition in oMSW, we use $\theta_{lt} \sim \mathcal{U}(\mathcal{S}_{\theta_{lt-1}}^{d-1})$ by first sampling $\theta'_{lt} \sim \mathcal{U}(\mathbb{S}^{d-1})$ then set $\theta_{lt} = \theta_{lt} - \frac{\langle \theta'_{lt}, \theta_{lt} \rangle}{\langle \theta'_{lt}, \theta'_{lt} \rangle} \theta'_{lt}$ then normalize $\theta_{lt} = \frac{\theta_{lt}}{||\theta_{lt}||_2}$. For deterministic input-awared transition, iMSW, we set $\theta_{lt} = \theta_{lt-1} + \eta \nabla_{\theta_{lt-1}} W_p(\theta_{lt-1} \sharp \mu, \theta_{lt-1} \sharp \nu)$ then normalize $\theta_{lt} = \frac{\theta_{lt}}{||\theta_{lt}||_2}$. For probabilistic input-awared transition, viMSW, $\theta_{lt} \sim \text{vMF}(\theta_t | \epsilon = \text{Proj}_{\mathbb{S}^{d-1}} \theta'_{lt}, \kappa)$ with $\theta'_{lt} = \theta_{lt-1} + \eta \nabla_{\theta_{lt-1}} W_p(\theta_{lt-1} \sharp \mu, \theta_{lt-1} \sharp \nu)$.

---

**Algorithm 6** Markovian sliced Wasserstein distance

---

**Input.** Probability measures $\mu, \nu$, the dimension $d$, the order $p$, the number of projections $L$, and the number of timesteps $T$.

**for** $l = 1$ to $L$ **do**

    Draw $\theta_{l0} \sim \sigma(\theta_0)$

    **for** $t = 1$ to $T - 1$ **do**

        Draw $\theta_{lt} \sim \sigma_t(\theta_t | \theta_{lt-1})$

    **end for**

**end for**

**Return.** $\left( \frac{1}{LT} \sum_{l=1}^{L} \sum_{t=1}^{T} W_p^p(\theta_{lt} \sharp \mu, \theta_{lt} \sharp \nu) \right)^{\frac{1}{p}}$

---

## A.4 Burned Thinned Markovian Sliced Wasserstein Distance

We continue the discussion on burned thinned MSW in Section 3.3. We first start with the Monte Carlo estimation of burned thinned MSW.

**Monte Carlo Estimation:** We samples $\theta_{11}, \ldots, \theta_{L1} \sim \sigma_1(\theta_1)$ for $L \geq 1$, then we samples $\theta_{lt} \sim \sigma_t(\theta_t | \theta_{lt-1})$ for $t = 1, \ldots, T$ and $l = 1, \ldots, L$. We then obtain samples $\theta'_{lt}$ by filtering out $t < M$ and $t\%N \neq 0$ from the set $\{\theta_{lt}\}$ for $l = 1, \ldots, L$ and $t = 1, \ldots, T$. The Monte Carlo approximation of the burned-thinned Markovian sliced Wasserstein distance is:

$$\widehat{\text{MSW}}_{p,T,N,M}(\mu, \nu) = \left( \frac{N}{L(T-M)} \sum_{l=1}^{L} \sum_{t=1}^{(T-M)/N} W_p^p \left( \theta'_{lt} \sharp \mu, \theta'_{lt} \sharp \nu \right) \right)^{\frac{1}{p}}. \tag{10}$$

**Theoretical properties.** We first state the following assumption: **A2.** Given $T > M \geq 0$, $N \geq 1$, the prior distribution $\sigma_1(\theta_1)$ and the transition distribution $\sigma_t(\theta_t | \theta_{t-1})$ are chosen such that there exists marginals $\sigma_t(\theta_t) = \int_{t^-} \sigma(\theta_1, \ldots, \theta_t) dt^-$ with $t \geq M$ and $t\%N = 0$, $t^- = \{t' = 1, \ldots, T | t' \neq t\}$.

The assumption **A2** can be easily obtained by using vMF transition, e.g., in probabilistic input-awared transition. From this assumption, we can derive theoretical properties of burned-thinned MSW including topological properties and statistical complexity.

**Proposition 4.** *For any $p \geq 1$, $T \geq 1$, $M \geq 0$, $N \geq 1$, and dimension $d \geq 1$, if **A2** holds, the burned thinned Markovian sliced Wasserstein distance $MSW_{p,T,N,M}(\cdot, \cdot)$ is a valid metric on the space of probability measures $\mathcal{P}_p(\mathbb{R}^d)$, namely, it satisfies the (i) non-negativity, (ii) symmetry, (iii) triangle inequality, and (iv) identity.*

The proof of Proposition 4 follows directly the proof of Theorem 1 in Appendix B.1.

**Proposition 5** (Weak Convergence). *For any $p \geq 1$, $T \geq 1$, $M \geq 0$, $N \geq 1$, and dimension $d \geq 1$, if **A2** holds, the convergence of probability measures in $\mathcal{P}_p(\mathbb{R}^d)$ under the burned thinned Markovian sliced Wasserstein distance $MSW_{p,T,N,M}(\cdot, \cdot)$ implies weak convergence of probability measures and vice versa.*

The proof of Proposition 5 follows directly the proof of Theorem 2 in Appendix B.2.

**Proposition 6.** *For any $p \geq 1$ and dimension $d \geq 1$, for any $T \geq 1$, $M \geq 0$, $N \geq 1$ and $\mu, \nu \in \mathcal{P}_p(\mathbb{R}^d)$, $MSW_{p,T,N,M}(\mu,\nu) \leq Max\text{-}SW_p(\mu,\nu) \leq W_p(\mu,\nu)$.*

The proof of Proposition 6 follows directly the proof of Proposition 1 in Appendix B.3.

**Proposition 7** (Sample Complexity). *Let $X_1, X_2, \ldots, X_n$ be i.i.d. samples from the probability measure $\mu$ being supported on compact set of $\mathbb{R}^d$. We denote the empirical measure $\mu_n = \frac{1}{n}\sum_{i=1}^{n}\delta_{X_i}$. Then, for any $p \geq 1$ and $T \geq 1$, $M \geq 0$, $N \geq 1$, there exists a universal constant $C > 0$ such that*

$$\mathbb{E}[MSW_{p,T,N,M}(\mu_n,\mu)] \leq C\sqrt{(d+1)\log n/n},$$

*where the outer expectation is taken with respect to the data $X_1, X_2, \ldots, X_n$.*

The proof of Proposition 7 follows directly the proof of Proposition 2 in Appendix B.4.

**Proposition 8** (Monte Carlo error). *For any $p \geq 1$, $T \geq 1$, $M \geq 0$, $N \geq 1$, dimension $d \geq 1$, and $\mu, \nu \in \mathcal{P}_p(\mathbb{R}^d)$, we have:*

$$\mathbb{E}|\widehat{MSW}^p_{p,T,N,M}(\mu,\nu) - MSW^p_{p,T,N,M}(\mu,\nu)|$$

$$\leq \frac{N}{\sqrt{L}T(T-M)}Var\left[\sum_{t=1}^{(T-M)/N} W_p^p\left(\theta_t'\sharp\mu, \theta_t'\sharp\nu\right)\right]^{\frac{1}{2}},$$

*where the variance is with respect to $\sigma(\theta_1', \ldots, \theta_{(T-M)/N}')$.*

The proof of Proposition 8 follows directly the proof of Proposition 3 in Appendix B.5.

## A.5  Discussions on Related Works

**K-SW is autoregressive decomposition.**   In MSW, we assume that the joint distribution over projecting directions has the first-order Markov structure: $\sigma(\theta_1, \ldots, \theta_T) = \sigma_1(\theta_1)\prod_{t=2}^{T}\sigma_t(\theta_t|\theta_{t-1})$. However, we can consider the full autoregressive decomposition $\sigma(\theta_1, \ldots, \theta_T) = \sigma_1(\theta_1)\prod_{t=2}^{T}\sigma_t(\theta_t|\theta_1, \ldots, \theta_{t-1})$. Let $T = K$ in K-SW, hence the transition distribution that is used in K-SW is: $\sigma_t(\theta_t|\theta_1, \ldots, \theta_{t-1}) = \text{Gram-Schmidt}_{\theta_1,\ldots,\theta_{t-1}}\sharp\mathcal{U}(\mathbb{S}^{d-1})$, where Gram-Schmidt$_{\theta_1,\ldots,\theta_{t-1}}(\theta_t)$ denotes the Gram-Schmidt process update that applies on $\theta_t$.

**Generalization of Max-K-SW.** Similar to Max-SW, we can derive a Markovian-based K-sliced Wasserstein distance that generalizes the idea of the projected gradient ascent update in Max-K-SW. However, the distance considers the transition on the Stiefel manifold instead of the unit hypersphere, hence, it will be more computationally expensive. Moreover, orthogonality might not be a good constraint. Therefore, the generalization of Max-K-SW might not have many advantages.

**Beyond the projected sub-gradient ascent update.** In the input-aware transition for MSW, we utilize the projected sub-gradient update as the transition function to create a new projecting direction. Therefore, we could other optimization techniques such as momentum, adaptive stepsize, and so on to create the transition function. We will leave the investigation about this direction to future work.

**Applications to other sliced Wasserstein variants.** The Markovian approach can be applied to other variants of sliced Wasserstein distances e.g., generalized sliced Wasserstein [26], augmented sliced Wasserstein distance [10], projected robust Wasserstein (PRW) [47, 32, 22] ($k > 1$ dimensional projection), convolution sliced Wasserstein [43], sliced partial optimal transport [6, 2], and so on.

**Markovian sliced Wasserstein distances in other applications.** We can apply MSW to the setting in [31] which is an implementation technique that utilizes both RAM and GPUs' memory for training sliced Wasserstein generative models. MSW can also replace sliced Wasserstein distance in pooling in [38]. Similarly, MSW can be used in applications that exist sliced Wasserstein distance e.g., clustering [28], Bayesian inference [39, 61], domain adaptation [60], and so on.

## B Proofs

### B.1 Proof of Theorem 1

**(i), (ii)**: the MSW is an expectation of the one-dimensional Wasserstein distance hence the non-negativity and symmetry properties of the MSW follow directly by the non-negativity and symmetry of the Wasserstein distance.

**(iii)** From the definition of MSW in Definition 1, given three probability measures $\mu_1, \mu_2, \mu_3 \in \mathcal{P}_p(\mathbb{R}^d)$ we have:

$$
\begin{aligned}
\mathrm{MSW}_{p,T}(\mu_1, \mu_3) &= \left( \mathbb{E}_{(\theta_{1:T}) \sim \sigma(\theta_{1:T})} \left[ \frac{1}{T} \sum_{t=1}^{T} W_p^p \left( \theta_t \sharp \mu_1, \theta_t \sharp \mu_3 \right) \right] \right)^{\frac{1}{p}} \\
&\leq \left( \mathbb{E}_{(\theta_{1:T}) \sim \sigma(\theta_{1:T})} \left[ \frac{1}{T} \sum_{t=1}^{T} \left( W_p \left( \theta_t \sharp \mu_1, \theta_t \sharp \mu_2 \right) + W_p \left( \theta_t \sharp \mu_2, \theta_t \sharp \mu_3 \right) \right)^p \right] \right)^{\frac{1}{p}} \\
&\leq \left( \mathbb{E}_{(\theta_{1:T}) \sim \sigma(\theta_{1:T})} \left[ \frac{1}{T} \sum_{t=1}^{T} W_p^p \left( \theta_t \sharp \mu_1, \theta_t \sharp \mu_2 \right) \right] \right)^{\frac{1}{p}} \\
&\quad + \left( \mathbb{E}_{(\theta_{1:T}) \sim \sigma(\theta_{1:T})} \left[ \frac{1}{T} \sum_{t=1}^{T} W_p^p \left( \theta_t \sharp \mu_2, \theta_t \sharp \mu_3 \right) \right] \right)^{\frac{1}{p}} \\
&= \mathrm{MSW}_{p,T}(\mu_1, \mu_2) + \mathrm{MSW}_{p,T}(\mu_2, \mu_3),
\end{aligned}
$$

where the first inequality is due to the triangle inequality of Wasserstein distance and the second inequality is due to the Minkowski inequality. We complete the triangle inequality proof.

**(iv)** We need to show that $\mathrm{MSW}_{p,T}(\mu, \nu) = 0$ if and only if $\mu = \nu$. First, from the definition of MSW, we obtain directly $\mu = \nu$ implies $\mathrm{MSW}_{p,T}(\mu, \nu) = 0$. For the reverse direction, we use the same proof technique in [8]. If $\mathrm{MSW}_{p,T}(\mu, \nu) = 0$, we have $\int_{\mathbb{S}^{(d-1) \otimes T}} \frac{1}{T} \sum_{t=1}^{T} W_p \left( \theta_t \sharp \mu, \theta_t \sharp \nu \right) \mathrm{d}\sigma(\theta_{1:T}) = 0$. If **A1** holds, namely, the prior distribution $\sigma_1(\theta_1)$ is supported on all the unit-hypersphere or exists a transition distribution $\sigma_t(\theta_t | \theta_{t-1})$ is supported on all the unit-hypersphere, we have $W_p(\theta \sharp \mu, \theta \sharp \nu) = 0$ for all $\theta \in \mathbb{S}^{d-1}$ where $\boldsymbol{\sigma}$ denotes the prior or the transition distribution that satisfies the assumption **A1**. From the identity property of the Wasserstein distance, we obtain $\theta \sharp \mu = \theta \sharp \nu$ for $\boldsymbol{\sigma}$-a.e $\theta \in \mathbb{S}^{d-1}$. Therefore, for any $t \in \mathbb{R}$ and $\theta \in \mathbb{S}^{d-1}$, we have:

$$
\begin{aligned}
\mathcal{F}[\mu](t\theta) &= \int_{\mathbb{R}^d} e^{-it\langle \theta, x \rangle} d\mu(x) = \int_{\mathbb{R}} e^{-itz} d\theta \sharp \mu(z) = \mathcal{F}[\theta \sharp \mu](t) \\
&= \mathcal{F}[\theta \sharp \nu](t) = \int_{\mathbb{R}} e^{-itz} d\theta \sharp \nu(z) = \int_{\mathbb{R}^d} e^{-it\langle \theta, x \rangle} d\nu(x) = \mathcal{F}[\nu](t\theta),
\end{aligned}
$$

where $\mathcal{F}[\gamma](w) = \int_{\mathbb{R}^{d'}} e^{-i\langle w, x \rangle} d\gamma(x)$ denotes the Fourier transform of $\gamma \in \mathcal{P}(\mathbb{R}^{d'})$. By the injectivity of the Fourier transform, we obtain $\mu = \nu$ which concludes the proof.

### B.2 Proof of Theorem 2

Our goal is to show that for any sequence of probability measures $(\mu_k)_{k \in \mathbb{N}}$ and $\mu$ in $\mathcal{P}_p(\mathbb{R}^d)$, $\lim_{k \to +\infty} \mathrm{MSW}_{p,T}(\mu_k, \mu) = 0$ if and only if for any continuous and bounded function $f : \mathbb{R}^d \to \mathbb{R}$, $\lim_{k \to +\infty} \int f \, \mathrm{d}\mu_k = \int f \, \mathrm{d}\mu$. The proof follows the techniques in [41]. We first state the following lemma.

**Lemma 1.** *For any $T \geq 1$, and dimension $d \geq 1$, if **A1** holds and a sequence of probability measures $(\mu_k)_{k \in \mathbb{N}}$ satisfies $\lim_{k \to +\infty} MSW_{1,T}(\mu_k, \mu) = 0$ with $\mu$ in $\mathcal{P}(\mathbb{R}^d)$, there exists an increasing function $\phi : \mathbb{N} \to \mathbb{N}$ such that the subsequence $(\mu_{\phi(k)})_{k \in \mathbb{N}}$ converges weakly to $\mu$.*

*Proof.* We are given that $\lim_{k \to +\infty} \mathrm{MSW}_{1,T}(\mu_k, \mu) = 0$, therefore $\lim_{k \to \infty} \int_{\mathbb{S}^{(d-1) \otimes T}} \frac{1}{T} \sum_{t=1}^{T} W_1 \left( \theta_t \sharp \mu_k, \theta_t \sharp \mu \right) \mathrm{d}\sigma(\theta_{1:T}) = 0$. If **A1** holds, namely, the prior

distribution $\sigma_1(\theta_1)$ is supported on all the unit-hypersphere or exists a transition distribution $\sigma_t(\theta_t|\theta_{t-1})$ is supported on all the unit-hypersphere, we have

$$\lim_{k\to\infty} \int_{\mathbb{S}^{d-1}} W_1\left(\theta\sharp\mu_k, \theta\sharp\mu\right) \mathrm{d}\boldsymbol{\sigma}(\theta) = 0,$$

where $\boldsymbol{\sigma}$ denotes the prior or the transition distribution that satisfies the assumption **A1**. From Theorem 2.2.5 in [3], there exists an increasing function $\phi : \mathbb{N} \to \mathbb{N}$ such that $\lim_{k\to\infty} W_1(\theta\sharp\mu_{\phi(k)}, \theta\sharp\nu) = 0$ for $\boldsymbol{\sigma}$-a.e $\theta \in \mathbb{S}^{d-1}$. Since the Wasserstein distance of implies weak convergence [58], $\left(\theta\sharp\mu_{\phi(k)}\right)_{k\in\mathbb{N}}$ converges weakly to $\theta\sharp\mu$ for $\boldsymbol{\sigma}$-a.e $\theta \in \mathbb{S}^{d-1}$.

Let $\Phi_\mu = \int_{\mathbb{R}^d} e^{i\langle v,w\rangle} \mathrm{d}\mu(w)$ be the characteristic function of $\mu \in \mathcal{P}(\mathbb{R}^d)$, we have the weak convergence implies the convergence of characteristic function (Theorem 4.3 [24]): $\lim_{k\to\infty} \Phi_{\theta\sharp\mu_{\phi(k)}}(s) = \Phi_{\theta\sharp\mu}(s)$, $\quad \forall s \in \mathbb{R}$, for $\boldsymbol{\sigma}$-a.e $\theta \in \mathbb{S}^{d-1}$. Therefore, $\lim_{k\to\infty} \Phi_{\mu_{\phi(k)}}(z) = \Phi_\mu(z)$, for almost most every $z \in \mathbb{R}^d$.

For any $\gamma > 0$ and a continuous function $f : \mathbb{R}^d \to \mathbb{R}$ with compact support, we denote $f_\gamma(x) = f * g_\gamma(x) = \left(2\pi\gamma^2\right)^{-d/2} \int_{\mathbb{R}^d} f(x-z) \exp\left(-\|z\|^2/\left(2\gamma^2\right)\right) \mathrm{d}z$ where $g_\gamma$ is the density function of $\mathcal{N}(0, \gamma I_d)$. We have:

$$
\begin{aligned}
\int_{\mathbb{R}^d} f_\gamma(z)\mathrm{d}\mu_{\phi(k)}(z) &= \int_{\mathbb{R}^d}\int_{\mathbb{R}^d} f(w)g_\gamma(z-w)\mathrm{d}w\, \mathrm{d}\mu_{\phi(k)}(z)\\
&= \int_{\mathbb{R}^d}\int_{\mathbb{R}^d} f(w)\left(2\pi\gamma^2\right)^{-d/2}\exp(-\|z-w\|^2/(2\gamma^2))\mathrm{d}w\, \mathrm{d}\mu_{\phi(k)}(z)\\
&= \left(2\pi\gamma^2\right)^{-d/2}\int_{\mathbb{R}^d}\int_{\mathbb{R}^d} f(w)\int_{\mathbb{R}^d} e^{i\langle z-w,x\rangle}g_{1/\gamma}(x)\mathrm{d}x\, \mathrm{d}w\, \mathrm{d}\mu_{\phi(k)}(z)\\
&= \left(2\pi\gamma^2\right)^{-d/2}\int_{\mathbb{R}^d}\int_{\mathbb{R}^d} f(w)\int_{\mathbb{R}^d} e^{-i\langle w,x\rangle}e^{i\langle z,x\rangle}g_{1/\gamma}(x)\mathrm{d}x\, \mathrm{d}w\, \mathrm{d}\mu_{\phi(k)}(z)\\
&= \left(2\pi\gamma^2\right)^{-d/2}\int_{\mathbb{R}^d}\int_{\mathbb{R}^d} f(w)e^{-i\langle w,x\rangle}g_{1/\gamma}(x)\int_{\mathbb{R}^d} e^{i\langle z,x\rangle}\mathrm{d}\mu_{\phi(k)}(z)\mathrm{d}x\, \mathrm{d}w\\
&= \left(2\pi\gamma^2\right)^{-d/2}\int_{\mathbb{R}^d}\int_{\mathbb{R}^d} f(w)e^{-i\langle w,x\rangle}g_{1/\gamma}(x)\Phi_{\mu_{\phi(k)}}(x)\mathrm{d}x\, \mathrm{d}w\\
&= \left(2\pi\gamma^2\right)^{-d/2}\int_{\mathbb{R}^d} \mathcal{F}[f](x)g_{1/\gamma}(x)\Phi_{\mu_{\phi(k)}}(x)\mathrm{d}x,
\end{aligned}
$$

where the third equality is due to the fact that $\int_{\mathbb{R}^d} e^{i\langle z-w,x\rangle}g_{1/\gamma}(x)\mathrm{d}x = \exp(-\|z-w\|^2/(2\gamma^2))$ and $\mathcal{F}[f](w) = \int_{\mathbb{R}^{d'}} f(x)e^{-i\langle w,x\rangle}dx$ denotes the Fourier transform of the bounded function $f$. Similarly, we have

$$
\begin{aligned}
\int_{\mathbb{R}^d} f_\gamma(z)\mathrm{d}\mu(z) &= \int_{\mathbb{R}^d}\int_{\mathbb{R}^d} f(w)g_\gamma(z-w)\mathrm{d}w\, \mathrm{d}\mu(z)\\
&= \int_{\mathbb{R}^d}\int_{\mathbb{R}^d} f(w)\left(2\pi\gamma^2\right)^{-d/2}\exp(-\|z-w\|^2/(2\gamma^2))\mathrm{d}w\, \mathrm{d}\mu(z)\\
&= \left(2\pi\gamma^2\right)^{-d/2}\int_{\mathbb{R}^d}\int_{\mathbb{R}^d} f(w)\int_{\mathbb{R}^d} e^{i\langle z-w,x\rangle}g_{1/\gamma}(x)\mathrm{d}x\, \mathrm{d}w\, \mathrm{d}\mu(z)\\
&= \left(2\pi\gamma^2\right)^{-d/2}\int_{\mathbb{R}^d}\int_{\mathbb{R}^d} f(w)\int_{\mathbb{R}^d} e^{-i\langle w,x\rangle}e^{i\langle z,x\rangle}g_{1/\gamma}(x)\mathrm{d}x\, \mathrm{d}w\, \mathrm{d}\mu(z)\\
&= \left(2\pi\gamma^2\right)^{-d/2}\int_{\mathbb{R}^d}\int_{\mathbb{R}^d} f(w)e^{-i\langle w,x\rangle}g_{1/\gamma}(x)\int_{\mathbb{R}^d} e^{i\langle z,x\rangle}\mathrm{d}\mu(z)\mathrm{d}x\, \mathrm{d}w\\
&= \left(2\pi\gamma^2\right)^{-d/2}\int_{\mathbb{R}^d}\int_{\mathbb{R}^d} f(w)e^{-i\langle w,x\rangle}g_{1/\gamma}(x)\Phi_\mu(x)\mathrm{d}x\, \mathrm{d}w\\
&= \left(2\pi\gamma^2\right)^{-d/2}\int_{\mathbb{R}^d} \mathcal{F}[f](x)g_{1/\gamma}(x)\Phi_\mu(x)\mathrm{d}x.
\end{aligned}
$$

Since $f$ is assumed to have compact support, $\mathcal{F}[f]$ exists and is bounded by $\int_{\mathbb{R}^d}|f(w)|\mathrm{d}w < +\infty$. Hence, for any $k \in \mathbb{R}$ and $x \in \mathbb{R}^d$, we have $\left|\mathcal{F}[f](x)g_{1/\gamma}(x)\Phi_{\mu_{\phi(k)}}(x)\right| \leq$

$g_{1/\gamma}(x) \int_{\mathbb{R}^d} |f(w)| \mathrm{d}w$ and $\left| \mathcal{F}[f](x)g_{1/\gamma}(x)\Phi_\mu(x) \right| \leq g_{1/\gamma}(x) \int_{\mathbb{R}^d} |f(w)| \mathrm{d}w$. Using the proved result of $\lim_{k\to\infty} \Phi_{\mu_{\phi(k)}}(z) = \Phi_\mu(z)$ and Lebesgue's Dominated Convergence Therefore, we obtain

$$
\begin{aligned}
\lim_{k\to\infty} \int_{\mathbb{R}^d} f_\gamma(z)\mathrm{d}\mu_{\phi(k)}(z) &= \lim_{k\to\infty} \left(2\pi\gamma^2\right)^{-d/2} \int_{\mathbb{R}^d} \mathcal{F}[f](x)g_{1/\gamma}(x)\Phi_{\mu_{\phi(k)}}(x)\mathrm{d}x \\
&= \left(2\pi\gamma^2\right)^{-d/2} \int_{\mathbb{R}^d} \mathcal{F}[f](x)g_{1/\gamma}(x)\Phi_{\mu_{\phi(k)}}(x)\mathrm{d}x \\
&= \int_{\mathbb{R}^d} f_\gamma(z)\mathrm{d}\mu(z).
\end{aligned}
$$

Moreover, we have:

$$
\begin{aligned}
&\lim_{\gamma\to 0} \limsup_{k\to+\infty} \left| \int_{\mathbb{R}^d} f(z)\mathrm{d}\mu_{\phi(k)}(z) - \int_{\mathbb{R}^d} f(z)\mathrm{d}\mu(z) \right| \\
&\leq \lim_{\gamma\to 0} \limsup_{k\to+\infty} \left[ 2\sup_{z\in\mathbb{R}^d} |f(z) - f_\gamma(z)| + \left| \int_{\mathbb{R}^d} f_\gamma(z)\mathrm{d}\mu_{\phi(k)}(z) - \int_{\mathbb{R}^d} f_\gamma(z)\mathrm{d}\mu(z) \right| \right] \\
&= \lim_{\gamma\to 0} 2\sup_{z\in\mathbb{R}^d} |f(z) - f_\gamma(z)| = 0,
\end{aligned}
$$

which implies $\left(\mu_{\phi(k)}\right)_{k\in\mathbb{N}}$ converges weakly to $\mu$. $\qquad\square$

We now continue the proof of Theorem 2. We first show that if $\lim_{k\to\infty} \mathrm{MSW}_{p,T}(\mu_k,\mu) = 0$, $(\mu_k)_{k\in\mathbb{N}}$ converges weakly to $\mu$. We consider a sequence $\left(\mu_{\phi(k)}\right)_{k\in\mathbb{N}}$ such that $\lim_{k\to\infty} \mathrm{MSW}_{p,T}(\mu_k,\mu) = 0$ and we suppose $\left(\mu_{\phi(k)}\right)_{k\in\mathbb{N}}$ does not converge weakly to $\mu$. Therefore, let $d_{\mathcal{P}}$ be the Lévy-Prokhorov metric, $\lim_{k\to\infty} d_{\mathcal{P}(\mu_k,\mu)} \neq 0$ that implies there exists $\varepsilon > 0$ and a subsequence $\left(\mu_{\psi(k)}\right)_{k\in\mathbb{N}}$ with an increasing function $\psi : \mathbb{N} \to \mathbb{N}$ such that for any $k \in \mathbb{N}$: $d_{\mathcal{P}}(\mu_{\psi(k)},\mu) \geq \varepsilon$. However, we have

$$
\begin{aligned}
\mathrm{MSW}_{p,T}(\mu,\nu) &= \left( \mathbb{E}_{(\theta_{1:T})\sim\sigma(\theta_{1:T})} \left[ \frac{1}{T}\sum_{t=1}^T W_p^p\left(\theta_t\sharp\mu, \theta_t\sharp\nu\right) \right] \right)^{\frac{1}{p}} \\
&\geq \mathbb{E}_{(\theta_{1:T})\sim\sigma(\theta_{1:T})} \left[ \frac{1}{T}\sum_{t=1}^T W_p\left(\theta_t\sharp\mu, \theta_t\sharp\nu\right) \right] \\
&\geq \mathbb{E}_{(\theta_{1:T})\sim\sigma(\theta_{1:T})} \left[ \frac{1}{T}\sum_{t=1}^T W_1\left(\theta_t\sharp\mu, \theta_t\sharp\nu\right) \right] = \mathrm{MSW}_{1,T}(\mu,\nu),
\end{aligned}
$$

by the Holder inequality with $\mu,\nu \in \mathbb{P}_p(\mathbb{R}^d)$. Therefore, $\lim_{k\to\infty} \mathrm{MSW}_{1,T}(\mu_{\psi(k)},\mu) = 0$ which implies that there exists s a subsequence $\left(\mu_{\phi(\psi(k))}\right)_{k\in\mathbb{N}}$ with an increasing function $\phi : \mathbb{N} \to \mathbb{N}$ such that $\left(\mu_{\phi(\psi(k))}\right)_{k\in\mathbb{N}}$ converges weakly to $\mu$ by Lemma 1. Hence, $\lim_{k\to\infty} d_{\mathcal{P}}\left(\mu_{\phi(\psi(k))},\mu\right) = 0$ which contradicts our assumption. We conclude that if $\lim_{k\to\infty} \mathrm{MSW}_{p,T}(\mu_k,\mu) = 0$, $(\mu_k)_{k\in\mathbb{N}}$ converges weakly to $\mu$.

Now, we show that if $(\mu_k)_{k\in\mathbb{N}}$ converges weakly to $\mu$, we have $\lim_{k\to\infty} \mathrm{MSW}_{p,T}(\mu_k,\mu) = 0$. By the continuous mapping theorem, we obtain $(\theta\sharp\mu_k)_{k\in\mathbb{N}}$ converges weakly to $\theta\sharp\mu$ for any $\theta \in \mathbb{S}^{d-1}$. Since the weak convergence implies the convergence under the Wasserstein distance [58], we obtain $\lim_{k\to\infty} W_p(\theta\sharp\mu_k,\mu) = 0$. Moreover, the Wasserstein distance is also bounded, hence the bounded convergence theorem:

$$
\begin{aligned}
\lim_{k\to\infty} \mathrm{MSW}_{p,T}^p(\mu_k,\mu) &= \mathbb{E}_{(\theta_{1:T})\sim\sigma(\theta_{1:T})} \left[ \frac{1}{T}\sum_{t=1}^T W_p^p\left(\theta_t\sharp\mu_k, \theta_t\sharp\mu\right) \right] \\
&= \mathbb{E}_{(\theta_{1:T})\sim\sigma(\theta_{1:T})} \left[ \frac{1}{T}\sum_{t=1}^T 0 \right] = 0.
\end{aligned}
$$

By the continuous mapping theorem with function $x \to x^{1/p}$, we obtain $\lim_{k\to\infty} \mathrm{MSW}_{p,T}(\mu_k,\mu) \to 0$ which completes the proof.

## B.3   Proof of Proposition 1

**(i)** We recall the definition of Max-SW:

$$\text{Max-SW}_p(\mu, \nu) = \max_{\theta \in \mathbb{S}^{d-1}} W_p(\theta \sharp \mu, \theta \sharp \nu).$$

Let $\theta^* = \text{argmax}_{\theta \in \mathbb{S}^{d-1}} W_p(\theta \sharp \mu, \theta \sharp \nu)$, from Definition 1, for any $p \geq 1$, $T \geq 1$, dimension $d \geq 1$, and $\mu, \nu \in \mathcal{P}_p(\mathbb{R}^d)$ we have:

$$\text{MSW}_{p,T}(\mu, \nu) = \left( \mathbb{E}_{(\theta_{1:T}) \sim \sigma(\theta_{1:T})} \left[ \frac{1}{T} \sum_{t=1}^{T} W_p^p \left( \theta_t \sharp \mu, \theta_t \sharp \nu \right) \right] \right)^{\frac{1}{p}}$$

$$\leq \left( \frac{1}{T} \sum_{t=1}^{T} W_p^p \left( \theta^* \sharp \mu, \theta^* \sharp \nu \right) \right)^{\frac{1}{p}} = W_p \left( \theta^* \sharp \mu, \theta^* \sharp \nu \right) = \text{Max-SW}_p(\mu, \nu).$$

Furthermore, by applying the Cauchy-Schwartz inequality, we have:

$$\text{Max-SW}_p^p(\mu, \nu) = \max_{\theta \in \mathbb{S}^{d-1}} \left( \inf_{\pi \in \Pi(\mu, \nu)} \int_{\mathbb{R}^d} \left| \theta^\top x - \theta^\top y \right|^p d\pi(x, y) \right)$$

$$\leq \max_{\theta \in \mathbb{S}^{d-1}} \left( \inf_{\pi \in \Pi(\mu, \nu)} \int_{\mathbb{R}^d \times \mathbb{R}^d} \|\theta\|^p \|x - y\|^p d\pi(x, y) \right)$$

$$= \inf_{\pi \in \Pi(\mu, \nu)} \int_{\mathbb{R}^d \times \mathbb{R}^d} \|\theta\|^p \|x - y\|^p d\pi(x, y)$$

$$= \inf_{\pi \in \Pi(\mu, \nu)} \int_{\mathbb{R}^d \times \mathbb{R}^d} \|x - y\|^p d\pi(x, y)$$

$$= W_p^p(\mu, \nu),$$

which completes the proof.

**(ii)** This result can be directly obtained from the definitions of MSW and SW.

## B.4   Proof of Proposition 2

In this proof, we denote $\Theta \subset \mathbb{R}^d$ as the compact set of the probability measure $\mu$. From Proposition 1, we find that

$$\mathbb{E}[\text{MSW}_{p,T}(\mu_n, \mu)] \leq \mathbb{E}\left[ \text{Max-SW}_p(\mu_n, \mu) \right].$$

Therefore, the proposition follows as long as we can demonstrate that

$$\mathbb{E}[\text{Max-SW}_p(\mu_n, \mu)] \leq C \sqrt{(d+1) \log_2 n / n}$$

where $C > 0$ is some universal constant and the outer expectation is taken with respect to the data. The proof for this result follows from the proof of Proposition 3 in [43]. Here, we provide the proof for the completeness. By defining $F_{n,\theta}$ and $F_\theta$ as the cumulative distributions of $\theta \sharp \mu_n$ and $\theta \sharp \mu$, the closed-form expression of the Wasserstein distance in one dimension leads to the following equations and inequalities:

$$\text{Max-SW}_p^p(\mu_n, \mu) = \max_{\theta \in \mathbb{S}^{d-1}} \int_0^1 |F_{n,\theta}^{-1}(u) - F_\theta^{-1}(u)|^p du$$

$$= \max_{\theta \in \mathbb{R}^d : \|\theta\|=1} \int_0^1 |F_{n,\theta}^{-1}(u) - F_\theta^{-1}(u)|^p du$$

$$\leq \text{diam}(\Theta) \max_{\theta \in \mathbb{R}^d : \|\theta\| \leq 1} |F_{n,\theta}(x) - F_\theta(x)|^p.$$

We can check that

$$\max_{\theta \in \mathbb{R}^d : \|\theta\| \leq 1} |F_{n,\theta}(x) - F_\theta(x)| = \sup_{B \in \mathcal{B}} |\mu_n(B) - \mu(B)|,$$

where $\mathcal{B}$ is the set of half-spaces $\{z \in \mathbb{R}^d : \theta^\top z \leq x\}$ for all $\theta \in \mathbb{R}^d$ such that $\|\theta\| \leq 1$. From VC inequality (Theorem 12.5in [16]), we have

$$\mathbb{P}\left(\sup_{B \in \mathcal{B}} |\mu_n(B) - \mu(B)| > t\right) \leq 8S(\mathcal{B}, n)e^{-nt^2/32}.$$

with $S(\mathcal{B}, n)$ is the growth function which is upper bounded by $(n+1)^{VC(\mathcal{B})}$ due to the Sauer Lemma (Proposition 4.18 in [59]). From Example 4.21 in [59], we get $VC(\mathcal{B}) = d + 1$.

Let $8S(\mathcal{B}, n)e^{-nt^2/32} \leq \delta$, we have $t^2 \geq \frac{32}{n} \log\left(\frac{8S(\mathcal{B}, n)}{\delta}\right)$. Therefore, we obtain

$$\mathbb{P}\left(\sup_{B \in \mathcal{B}} |\mu_n(B) - \mu(B)| \leq \sqrt{\frac{32}{n} \log\left(\frac{8S(\mathcal{B}, n)}{\delta}\right)}\right) \geq 1 - \delta,$$

Now we convert the probability statement to inequality with expectation. Using the Jensen inequality and the tail sum expectation for non-negative random variable, we have:

$$\mathbb{E}\left[\sup_{B \in \mathcal{B}} |\mu_n(B) - \mu(B)|\right]$$

$$\leq \sqrt{\mathbb{E}\left[\sup_{B \in \mathcal{B}} |\mu_n(B) - \mu(B)|\right]^2} = \sqrt{\int_0^\infty \mathbb{P}\left(\left(\sup_{B \in \mathcal{B}} |\mu_n(B) - \mu(B)|\right)^2 > t\right) dt}$$

$$= \sqrt{\int_0^u \mathbb{P}\left(\left(\sup_{B \in \mathcal{B}} |\mu_n(B) - \mu(B)|\right)^2 > t\right) dt + \int_u^\infty \mathbb{P}\left(\left(\sup_{B \in \mathcal{B}} |\mu_n(B) - \mu(B)|\right)^2 > t\right) dt}$$

$$\leq \sqrt{\int_0^u 1 dt + \int_u^\infty 8S(\mathcal{B}, n)e^{-nt/32} dt} = \sqrt{u + 256S(\mathcal{B}, n)\frac{e^{-nu/32}}{n}}.$$

Let $f(u) = u + 256S(\mathcal{B}, n)\frac{e^{-nu/32}}{n}$, we have $f'(u) = 1 + 8S(\mathcal{B}, n)e^{-nu/32}$, hence the minima $u^\star = \frac{32 \log(8S(\mathcal{B}, n))}{n}$. Since the inequality holds for any $u$, we have:

$$\mathbb{E}\left[\sup_{B \in \mathcal{B}} |\mu_n(B) - \mu(B)|\right] \leq \sqrt{\frac{32 \log(8S(\mathcal{B}, n))}{n} + 32} \leq C\sqrt{\frac{(d+1) \log(n+1)}{n}},$$

by using Sauer Lemma. Putting the above results together leads to

$$\mathbb{E}[\text{Max-SW}_p(\mu_n, \mu)] \leq C\sqrt{(d+1) \log_2 n / n},$$

where $C > 0$ is some universal constant. As a consequence, we obtain the conclusion of the proposition.

## B.5 Proof of Proposition 3

For any $p \geq 1$, $T \geq 1$, dimension $d \geq 1$, and $\mu, \nu \in \mathcal{P}_p(\mathbb{R}^d)$, using the Holder's inequality, we have:

$$\mathbb{E}|\widehat{\text{MSW}}_{p,T}^p(\mu, \nu) - \text{MSW}_{p,T}^p(\mu, \nu)|$$

$$\leq \left(\mathbb{E}|\widehat{\text{MSW}}_{p,T}^p(\mu, \nu) - \text{MSW}_{p,T}^p(\mu, \nu)|^2\right)^{\frac{1}{2}}$$

$$= \left(\mathbb{E}\left|\frac{1}{TL}\sum_{t=1}^T\sum_{l=1}^L W_p^p(\theta_{tl}\sharp\mu, \theta_{tl}\sharp\nu) - \mathbb{E}_{\theta_{1:T}\sim\sigma(\theta_{1:T})}\left[\frac{1}{T}\sum_{t=1}^T W_p^p(\theta_t\sharp\mu, \theta_t\sharp\nu)\right]\right|^2\right)^{\frac{1}{2}}$$

$$= \left(Var\left[\frac{1}{TL}\sum_{t=1}^T\sum_{l=1}^L W_p^p(\theta_{tl}\sharp\mu, \theta_{tl}\sharp\nu)\right]\right)^{\frac{1}{2}}$$

$$= \frac{1}{T\sqrt{L}}Var\left[\sum_{t=1}^T W_p^p(\theta_t\sharp\mu, \theta_t\sharp\nu)\right]^{\frac{1}{2}},$$

which completes the proof.

**Algorithm 7** Gradient flow with the Euler scheme

---

**Input.** the start distribution $\mu = \frac{1}{n}\sum_{i=1}^{n}\delta_{X_i}$, the target distribution $\nu = \frac{1}{n}\sum_{i=1}^{n}\delta_{Y_i}$, number of Euler iterations $T$ (abuse of notation), Euler step size $\eta$ (abuse of notation), a metric $\mathcal{D}$.
**for** $t = 1$ to $T$ **do**
  $X = X - n \cdot \eta \nabla_X \mathcal{D}(P_X, P_Y)$
**end for**
**Output.** $\mu = \frac{1}{n}\sum_{i=1}^{n}\delta_{X_i}$

---

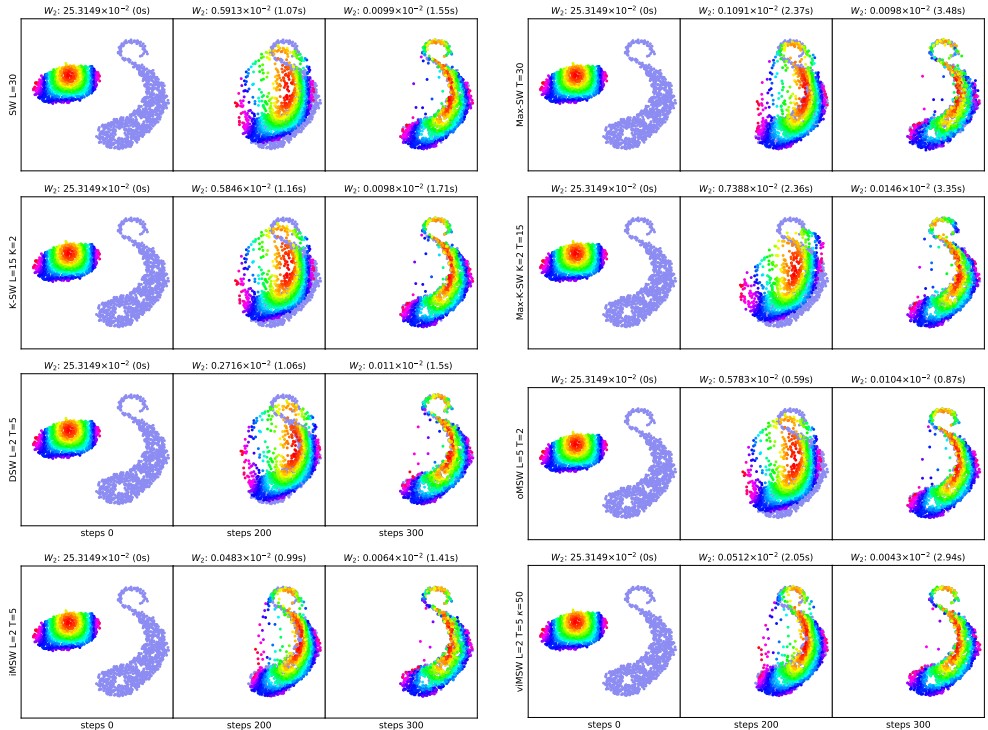

Figure 4: The figures show the gradient flows that are from the empirical distribution over the color points to the empirical distribution over S-shape points produced by different distances. The corresponding Wasserstein-2 distance between the empirical distribution at the current step and the S-shape distribution and the computational time (in second) to reach the step is reported at the top of the figure.

## C  Additional Experiments

In this section, we present the detail of experimental frameworks and additional experiments on gradient flows, color transfer, and deep generative modeling which are not in the main paper.

### C.1  Gradient Flows

**Framework.** We have discussed in detail the framework of gradient flow in Section 4.1 in the main paper. Here, we summarize the Euler scheme for solving the gradient flow in Algorithm 7.

**Visualization of gradient flows.** We show the visualization of gradient flows from all distances (Table 1) in Figure 4. Overall, we observe that the quality of the flows is consistent with the quantitative Wasserstein-2 score which is computed using [18]. From the figures, we see that iMSW and viMSW help the flows converge very fast. Namely, Wasserstein-2 scores at steps 200 of iMSW and viMSW are much lower than other distances. For oMSW, with $L = 5, T = 2$, it achieves a comparable result to SW, K-SW, and Max-SW while being faster.

**Studies on hyper-parameters.** We run gradient flows with different values of hyper-parameters and report the Wasserstein-2 scores and computational time in Table 3. From the table and Figure 4, we

Table 3: Summary of Wasserstein-2 scores, computational time in second (s) of different distances in gradient flow application.

| Distances | Wasserstein-2 ($\downarrow$) | Time ($\downarrow$) | Distances | Wasserstein-2 ($\downarrow$) | Time ($\downarrow$) |
|---|---|---|---|---|---|
| SW (L=10) | $0.0113 \times 10^{-2}$ | 0.85 | SW (L=100) | $0.0096 \times 10^{-2}$ | 4.32 |
| Max-SW (T=5) | $0.0231 \times 10^{-2}$ | 1.02 | Max-SW (T=100) | $0.0083 \times 10^{-2}$ | 10.46 |
| K-SW (L=5,K=2) | $0.0104 \times 10^{-2}$ | 0.92 | K-SW (L=20,K=2) | $0.0096 \times 10^{-2}$ | 1.97 |
| Max-K-SW (K=2,T=5) | $0.0152 \times 10^{-2}$ | 1.41 | Max-K-SW (K=2,T=100) | $0.0083 \times 10^{-2}$ | 10.46 |
| iMSW (L=1,T=5) | $0.0109 \times 10^{-2}$ | 1.07 | iMSW (L=5,T=5) | $0.0055 \times 10^{-2}$ | 2.44 |
| iMSW (L=2,T=10) | $0.0052 \times 10^{-2}$ | 2.79 | iMSW (L=5,T=2) | $0.0071 \times 10^{-2}$ | 1.14 |
| iMSW (L=2,T=5,M=4) | $0.0101 \times 10^{-2}$ | 1.2 | iMSW (L=2,T=5,M=2) | $0.0055 \times 10^{-2}$ | 1.25 |
| iMSW (L=2,T=5,M=0,N=2) | $0.0066 \times 10^{-2}$ | 1.28 | iMSW (L=2,T=5,M=2,N=2) | $0.0072 \times 10^{-2}$ | 1.19 |
| viMSW (L=2,T=5,$\kappa$=10) | $0.0052 \times 10^{-2}$ | 3.12 | viMSW (L=2,T=5,$\kappa$=100) | $0.0053 \times 10^{-2}$ | 2.76 |

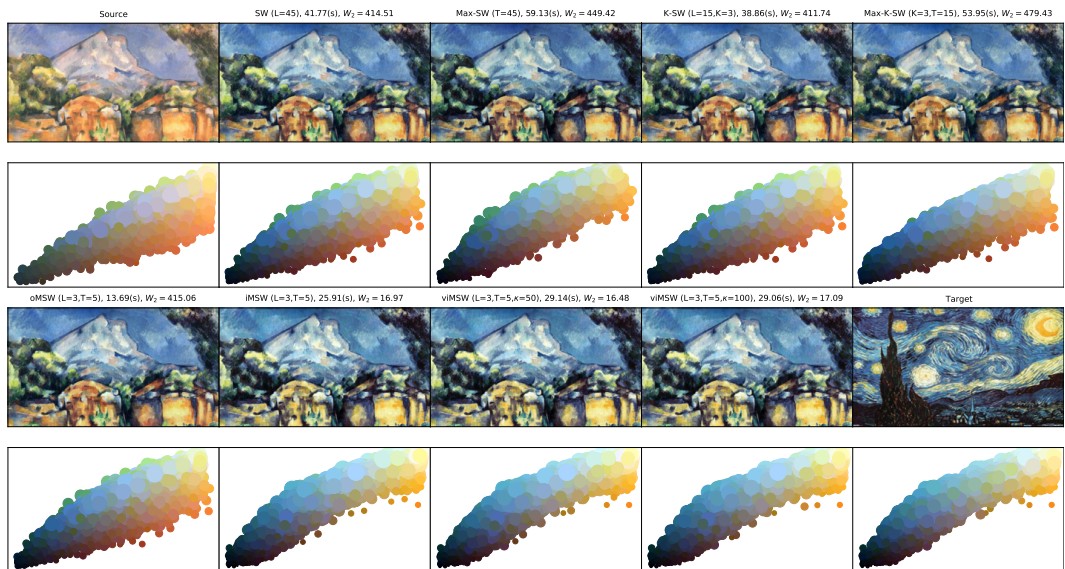

Figure 5: The figures show the source image, the target image, and transferred images from different distances. The corresponding Wasserstein-2 distance between the empirical distribution over transferred color palates and the empirical distribution over the target color palette and the computational time (in second) is reported at the top of the figure. The color palates are given below the corresponding images.

see that SW with $L = 10$ is worse than oMSW, iMSW, and viMSW with $L = 2, T = 5$ (10 total projections). Increasing the number of projections to 100, SW gets better, however, its Wasserstein-2 score is still higher than the scores of iMSW and viMSW while its computational time is bigger. Similarly, Max-(K)-SW with $T = 100$ is better than Max-(K)-SW with $T = 5$ and $T = 10$, however, it is still worse than iMSW and viMSW in terms of computation and performance. For burning and thinning, we see that the technique can help improve the computation considerably. More importantly, the burning and thinning techniques do not reduce the performance too much. For iMSW, increasing $L$ and $T$ leads to a better flow. For the same number of total projections e.g., 10, $L = 2, T = 5$ is better than $L = 5, T = 2$. For viMSW, it usually performs better than iMSW, however, its computation is worse due to the sampling complexity of the vMF distribution. We vary the concentration parameter $\kappa \in \{10, 50, 100\}$ and find that $\kappa = 50$ is the best. Hence, it might suggest that a good balance between heading to the "max" projecting direction and exploring the space of projecting directions is the best strategy.

## C.2   Color Transfer

**Framework.** In our experiments, we first compress the color palette of the source image and the target image to 3000 colors by using K-Mean clustering. After that, the color transfer application is

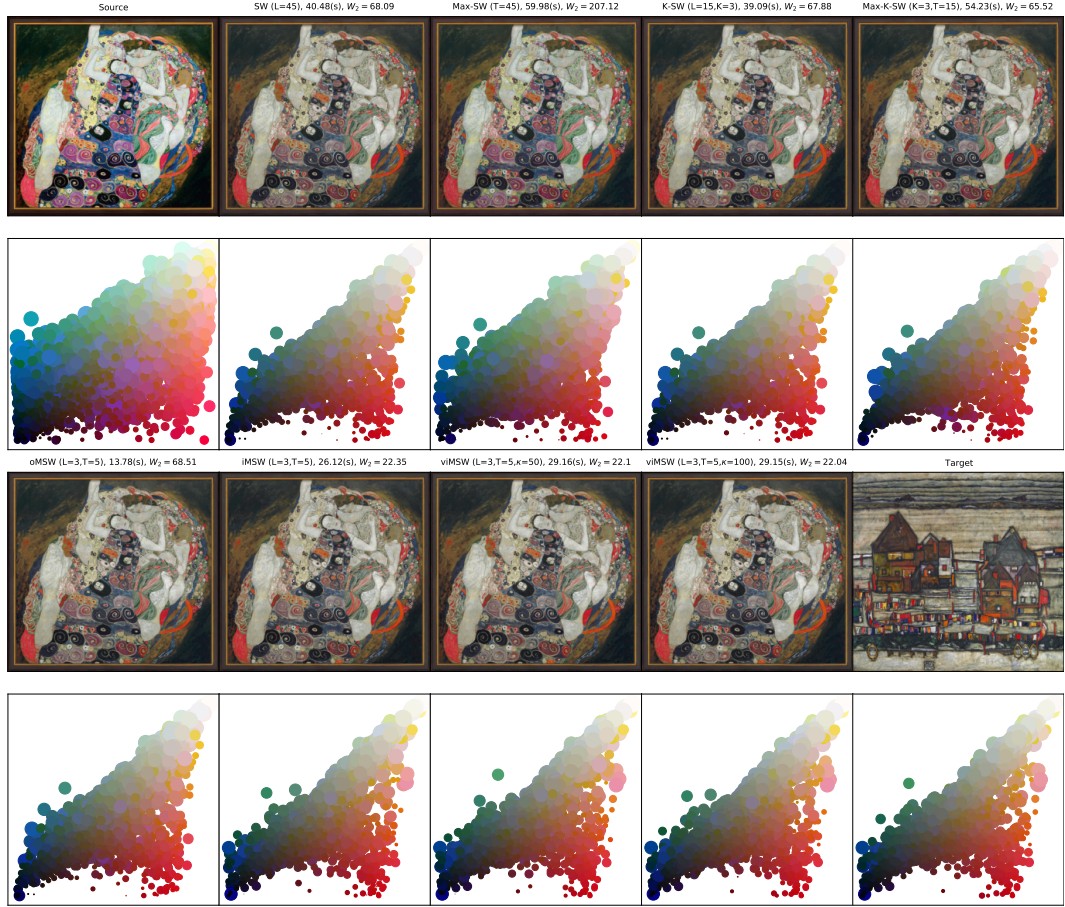

Figure 6: The figures show the source image, the target images, and transferred images from different distances. The corresponding Wasserstein-2 distance between the empirical distribution over transferred color palates and the empirical distribution over the target color palette and the computational time (in second) is reported at the top of the figure. The color palates are given below the corresponding images.

---

**Algorithm 8** Color Transfer

---

**Input.** source color palette $X \in \{0, \ldots, 255\}^{n \times 3}$, target color palette $Y \in \{0, \ldots, 255\}^{n \times 3}$, number of Euler iterations $T$ (abuse of notation), Euler step size $\eta$ (abuse of notation), a metric $\mathcal{D}$.
**for** $t = 1$ to $T$ **do**
    $X = X - n \cdot \eta \nabla_X \mathcal{D}(P_X, P_Y)$
**end for**
$X = \text{round}(X, \{0, \ldots, 255\})$
**Output.** $X$

---

conducted by using Algorithm 8 which is a modified version of the gradient flow algorithm since the color palette contains only positive integer in $\{0, \ldots, 255\}$. The flow can be seen as an incomplete transportation map that maps from the source color palette to a color palette that is close to the target color palette. This is quite similar to the iterative distribution transfer algorithm [8], however, the construction of the iterative map is different.

**Visuallization of transferred images.** We show the source image, the target image, and the corresponding transferred images from distances in Figure 5 and Figure 6. The color palates are given below the corresponding images. The corresponding Wasserstein-2 distance between the empirical distribution over transferred color palates and the empirical distribution over the target color palette and the computational time (in second) is reported at the top of the figure. First, we observe that

Table 4: Summary of Wasserstein-2 scores, computational time in second (s) of different distances in the color transfer application.

| Distances | Wasserstein-2 ($\downarrow$) | Time ($\downarrow$) | Distances | Wasserstein-2 ($\downarrow$) | Time ($\downarrow$) |
|---|---|---|---|---|---|
| SW (L=45) | 414.51 | 41.77 | SW (L=15) | 421.5 | 12.96 |
| Max-SW (T=45) | 449.42 | 59.13 | Max-SW (T=15) | 450.37 | 19.03 |
| K-SW (L=15,K=3) | 411.74 | 38.86 | K-SW (L=5,K=3) | 413.16 | 14.2 |
| Max-K-SW (K=3,T=15) | 479.43 | 53.95 | Max-K-SW (K=3,T=5) | 510.43 | 17.46 |
| oMSW (L=3,T=5) | 415.06 | 13.69 | oMSW (L=3,T=15) | 414.29 | 38.51 |
| iMSW (L=3,T=5) | 16.97 | 25.91 | iMSW (L=3,T=15) | 15.23 | 79.47 |
| iMSW (L=5,T=5) | 21.63 | 39.82 | iMSW (L=5,T=3) | 24.02 | 22.27 |
| iMSW (L=3,T=15,M=14) | 26.23 | 48.08 | iMSW (L=3,T=15,M=10) | 18.67 | 55.55 |
| iMSW (L=3,T=15,M=0,N=2) | 16.6 | 62.66 | iMSW (L=3,T=15,M=10,N=2) | 19.2 | 50.1 |
| viMSW (L=3,T=5,$\kappa$=50) | 16.48 | 29.14 | viMSW (L=3,T=5,$\kappa$=100) | 16.49 | 29.06 |

the qualitative comparison (transferred images and color palette) is consistent with the Wasserstein scores. We observe that iMSW and viMSW have their transferred images closer to the target image in terms of color than other distances. More importantly, iMSW and viMSW are faster than other distances. Max-SW and Max-K-SW do not perform well in this application, namely, they are slow and give high Wasserstein distances. For oMSW, it is comparable to SW and K-SW while being faster.

**Studies on hyper-parameters.** In addition to result in Figure 5, we run color transfer with other settings of distances in Table 4. From the table, increasing the number of projections $L$ lead to a better result for SW and K-SW. However, they are still worse than iMSW and viMSW with a smaller number of projections. Similarly, increasing $T$ helps Max-SW, Max-K-SW, and iMSW better. As discussed in the main paper, the burning and thinning technique improves the computation and sometimes enhances the performance.

## C.3  Deep Generative Models

**Framework.** We follow the generative modeling framework from [20, 42]. Here, we state an adaptive formulation of the framework. We are given a data distribution $\mu \in \mathcal{P}(\mathcal{X})$ through its random samples (data). Our goal is to estimate a parametric distribution $\nu_\phi$ that belongs to a family of distributions indexed by parameters $\phi$ in a parameter space $\Phi$. Deep generative modeling is interested in constructing $\nu_\phi$ via pushforward measure. In particular, $\nu_\phi$ is implicitly represented by pushing forward a random noise $\nu_0 \in \mathcal{P}(\mathcal{Z})$ e.g., standard multivariable Gaussian, through a parametric function $G_\phi : \mathcal{Z} \to \mathcal{X}$ (a neural network with weights $\phi$). To estimate $\phi$ ($\nu_\phi$), the expected distance estimator [54, 41] is used:

$$\text{argmin}_{\phi \in \Phi} \mathbb{E}_{(X,Z) \sim \mu^{\otimes m} \otimes \nu_0^{\otimes m}} [\mathcal{D}(P_X, P_{G_\phi(Z)})],$$

where $m \geq 1$, $\mathcal{D}$ can be any distance on space of probability measures, $\mu^{\otimes}$ is the product measures, namely, $X = (x_1, \ldots, x_m) \sim \mu^{\otimes}$ is equivalent to $x_i \sim \mu$ for $i = 1, \ldots, m$, and $P_X = \frac{1}{m} \sum_{i=1}^m \delta_{x_i}$. Similarly, $Z = (z_1, \ldots, z_m)$ with $z_i \sim \nu_0$ for $i = 1, \ldots, m$, and $G_\phi(Z)$ is the output of the neural work given the input mini-batch $Z$.

By using Wasserstein distance, sliced Wasserstein distance, and their variants as the distance $\mathcal{D}$, we obtain the corresponding estimators. However, applying directly those estimators to natural image data cannot give perceptually good results [20, 15]. The reason is that Wasserstein distance, sliced Wasserstein distances, and their variants require a ground metric as input e.g., $\mathcal{L}_2$, however, those ground metrics are not meaningful on images. Therefore, previous works propose using a function that maps the original data space $\mathcal{X}$ to a feature space $\mathcal{F}$ where the $\mathcal{L}_2$ norm is meaningful [52]. We denote the feature function $F_\gamma : \mathcal{X} \to \mathcal{F}$. Now the estimator becomes:

$$\text{argmin}_{\phi \in \Phi} \mathbb{E}_{(X,Z) \sim \mu^{\otimes m} \otimes \nu_0^{\otimes m}} [\mathcal{D}(P_{F_\gamma(X)}, P_{F_\gamma(G_\phi(Z))})].$$

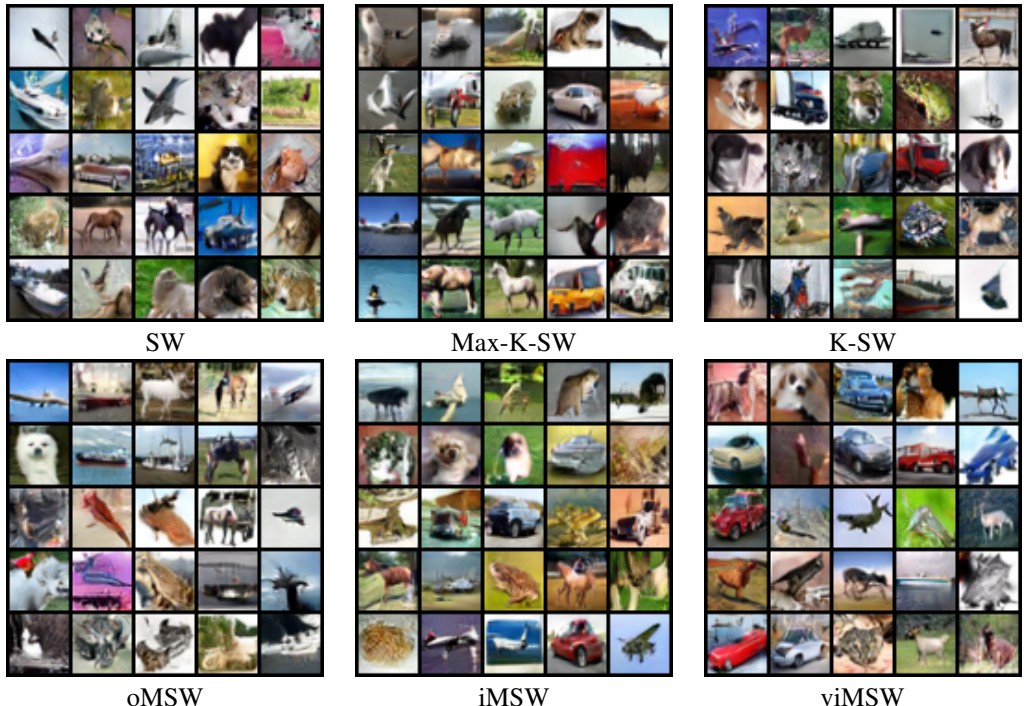



SW   Max-K-SW   K-SW

oMSW   iMSW   viMSW



Figure 7: Random generated images of distances on CIFAR10.

The above optimization can be solved by stochastic gradient descent algorithm with the following stochastic gradient estimator:

$$\nabla_\phi \mathbb{E}_{(X,Z)\sim\mu^{\otimes m}\otimes\nu_0^{\otimes m}}[\mathcal{D}(P_{F_\gamma(X)}, P_{F_\gamma(G_\phi(Z))})] = \mathbb{E}_{(X,Z)\sim\mu^{\otimes m}\otimes\nu_0^{\otimes m}}[\nabla_\phi \mathcal{D}(P_{F_\gamma(X)}, P_{F_\gamma(G_\phi(Z))})]$$

$$\approx \frac{1}{K}\sum_{k=1}^K \nabla_\phi \mathcal{D}(P_{F_\gamma(X_k)}, P_{F_\gamma(G_\phi(Z_k))}),$$

where $X_1, \ldots, X_K$ are drawn i.i.d from $\mu^{\otimes m}$ and $Z_1, \ldots, Z_K$ are drawn i.i.d from $\nu_0^{\otimes m}$. There are several ways to estimate the feature function $F_\gamma$ in practice. In our experiments, we use the following objective [15]:

$$\min_\gamma \left(\mathbb{E}_{X\sim\mu^{\otimes m}}[\min(0, -1 + H(F_\gamma(X)))] + \mathbb{E}_{Z\sim\nu_0^{\otimes m}}[\min(0, -1 - H(F_\gamma(G_\phi(Z))))]\right),$$

where $H : \mathcal{F} \to \mathbb{R}$. The above optimization problem is also solved by the stochastic gradient descent algorithm with the following gradient estimator:

$$\nabla_\gamma \left(\mathbb{E}_{X\sim\mu^{\otimes m}}[\min(0, -1 + H(F_\gamma(X)))] + \mathbb{E}_{Z\sim\nu_0^{\otimes m}}[\min(0, -1 - H(F_\gamma(G_\phi(Z))))]\right)$$

$$= \mathbb{E}_{X\sim\mu^{\otimes m}}[\nabla_\gamma \min(0, -1 + H(F_\gamma(X)))] + \mathbb{E}_{Z\sim\nu_0^{\otimes m}}[\nabla_\gamma \min(0, -1 - H(F_\gamma(G_\phi(Z))))]$$

$$\approx \frac{1}{K}\sum_{k=1}^K[\nabla_\gamma \min(0, -1 + H(F_\gamma(X_k)))] + \frac{1}{K}\sum_{k=1}^K[\nabla_\gamma \min(0, -1 - H(F_\gamma(G_\phi(Z_k))))],$$

where $X_1, \ldots, X_K$ are drawn i.i.d from $\mu^{\otimes m}$ and $Z_1, \ldots, Z_K$ are drawn i.i.d from $\nu_0^{\otimes m}$.

**Settings.** We use the following neural networks for $G_\phi$ and $F_\gamma$:

- **CIFAR10**:
    - $G_\phi$: $z \in \mathbb{R}^{128}(\sim \nu_0 : \mathcal{N}(0,1)) \to 4 \times 4 \times 256(\text{Dense, Linear}) \to$ ResBlock up 256 $\to$ ResBlock up 256 $\to$ ResBlock up 256 $\to$ BN, ReLU, $\to$ $3 \times 3$ conv, 3 Tanh .
    - $F_{\gamma_1}$: $x \in [-1,1]^{32\times32\times3} \to$ ResBlock down 128 $\to$ ResBlock down 128 $\to$ ResBlock down 128 $\to$ ResBlock 128 $\to$ ResBlock 128.

Table 5: Summary of FID and IS scores of methods on CIFAR10 (32x32), and CelebA (64x64)

| Method | CIFAR10 (32x32) | | CelebA (64x64) |
| | FID ($\downarrow$) | IS ($\uparrow$) | FID ($\downarrow$) |
|---|---|---|---|
| iMSW (L=100,T=10,M=0,N=1) | 14.61±0.72 | 8.15±0.15 | 9.73±0.33 |
| iMSW (L=100,T=10,M=9,N=1) | 14.16±1.11 | 8.17±0.07 | 9.10±0.34 |
| iMSW (L=100,T=10,M=5,N=1) | 13.93±0.21 | 8.15±0.05 | 9.49±0.52 |
| iMSW (L=100,T=10,M=0,N=2) | 14.33±0.32 | 8.15±0.06 | 8.99±0.64 |
| iMSW (L=10,T=100,M=0,N=1) | 14.26±0.74 | 8.15±0.07 | 8.89±0.23 |
| iMSW (L=10,T=100,M=99,N=1) | 14.50±0.70 | 8.12±0.08 | 9.55±0.35 |
| iMSW (L=10,T=100,M=50,N=1) | 14.41±0.58 | 8.12±0.06 | 9.46±0.73 |
| iMSW (L=10,T=100,M=0,N=2) | 14.65±0.01 | 8.11±0.06 | 9.49±0.39 |

- $F_{\gamma_2}$: $x \in \mathbb{R}^{128 \times 8 \times 8} \to$ ReLU $\to$ Global sum pooling(128) $\to$ 1(Spectral normalization).
- $F_\gamma(x) = (F_{\gamma_1}(x), F_{\gamma_2}(F_{\gamma_1}(x)))$ and $H(F_\gamma(x)) = F_{\gamma_2}(F_{\gamma_1}(x))$.

- **CelebA.**
  - $G_\phi$: $z \in \mathbb{R}^{128}(\sim \nu_0 : \mathcal{N}(0,1)) \to 4 \times 4 \times 256$(Dense, Linear) $\to$ ResBlock up 256 $\to$ ResBlock up 256 $\to$ ResBlock up 256 $\to$ ResBlock up 256 $\to$ BN, ReLU, $\to 3 \times 3$ conv, 3 Tanh .
  - $F_{\gamma_1}$: $x \in [-1,1]^{32 \times 32 \times 3} \to$ ResBlock down 128 $\to$ ResBlock down 128 $\to$ ResBlock down 128 $\to$ ResBlock 128 $\to$ ResBlock 128.
  - $F_{\gamma_2}$: $x \in \mathbb{R}^{128 \times 8 \times 8} \to$ ReLU $\to$ Global sum pooling(128) $\to$ 1(Spectral normalization).
  - $F_\gamma(x) = (F_{\gamma_1}(x), F_{\gamma_2}(F_{\gamma_1}(x)))$ and $H(F_\gamma(x)) = F_{\gamma_2}(F_{\gamma_1}(x))$.

For all datasets, the number of training iterations is set to 50000. We update the generator $G_\phi$ each 5 iterations while we update the feature function $F_\gamma$ every iteration. The mini-batch size $m$ is set 128 in all datasets. The learning rate for $G_\phi$ and $F_\gamma$ is 0.0002 and the optimizer is Adam [25] with parameters $(\beta_1, \beta_2) = (0, 0.9)$. We use the order $p = 2$ for all sliced Wasserstein variants. We use 50000 random samples from estimated generative models $G_\phi$ for computing the FID scores and the Inception scores. In evaluating FID scores, we use all training samples for computing statistics of datasets[2].

**Generated images.** We show generated images on CIFAR10 and CelebA from different generative models trained by different distances in Figure 7 and in Figure 8 in turn. Overall, images are visually consistent with the quantitative FID scores in Table 2.

**Studies on hyperparameters.** We run some additional settings of iMSW to investigate the performance of the burning thinning technique and to compare the role of $L$ and $T$ in Table 5. First, we see that burning and thinning helps to improve FID score and IS score on CIFAR10 and CelebA in the settings of $L = 100, T = 10$. It is worth noting that the original purpose of burning and thinning is to reduce computational complexity and memory complexity. The side benefit of improving performance requires more investigation that is left for future work. In addition, we find that for the same number of total projections 1000 without burning and thinning, the setting of $L = 10, T = 100$ is better than the setting of $L = 100, T = 10$ on CIFAR10. However, the reverse direction happens on CelebA. Therefore, on different datasets, it might require hyperparameter tunning for finding the best setting of the number of projections $L$ and the number of timesteps $T$.

---

[2]We evaluate the scores based on the code from `https://github.com/GongXinyuu/sngan.pytorch`.

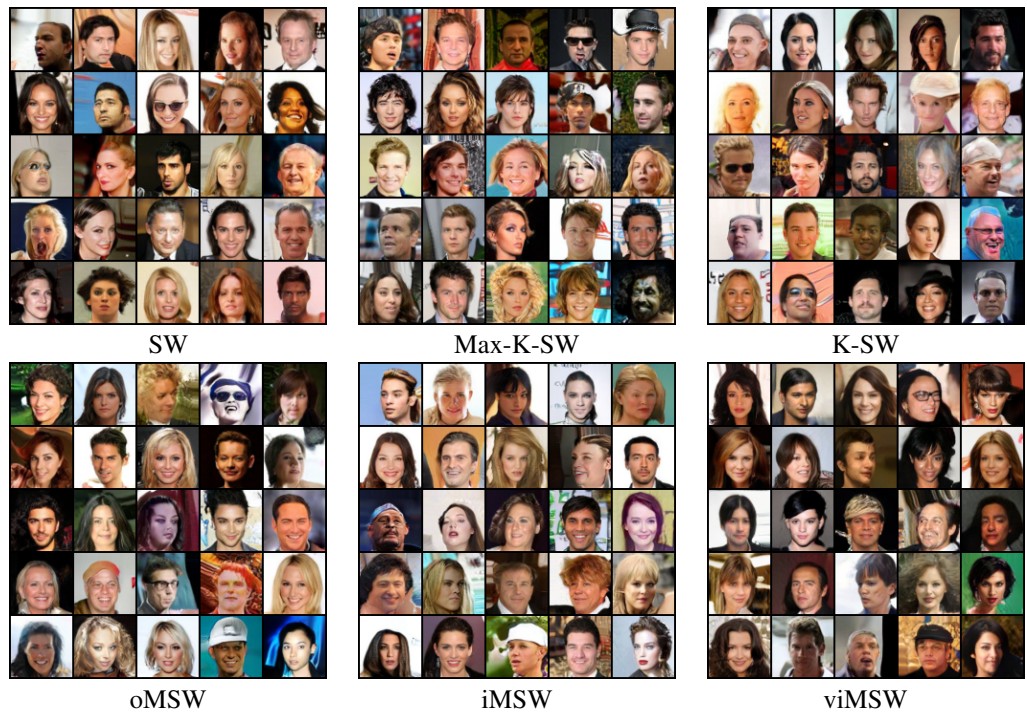

Figure 8: Random generated images of distances on CelebA.

