# OpenReview forum: "Markovian Sliced Wasserstein Distances: Beyond Independent Projections"
_NeurIPS.cc/2023/Conference — NeurIPS 2023 poster_

### Official Review · Reviewer_YkFo · 2023-06-19

**Soundness:** 3 good
**Presentation:** 3 good
**Contribution:** 3 good
**Rating:** 7
**Confidence:** 4

**Summary:**

The authors focus on the study of sliced Wasserstein distances.
It consists in projecting measures onto Euclidean lines, such that OT between these projected measures has low complexity because it amounts to sort samples.
The projections are unit vectors of the d-dimensional sphere of R^d, and in practice one samples a set of projections uniformly samples over the unit sphere.
However, this might be sub-efficient since some projections might be redundant by sampling close directions, or uniformative by projecting in directions where measures are similar.

To solve this issue, the authors propose to sample projections non-uniformly. In particular they consider sampling processus satisfying a first-order Markov property.
They propose a sliced OT variant called 'Markovian Sliced Wasserstein'.
They prove metric, weak* convergence and statistical properties.
They propose two families of Markov processes, focusing on orthogonality or data-driven.
They propose a 'burning and thinning' variant to reduce the number of projections and improve complexity.
They run experiments of gradient flows, color transfer and generative learning to illustrate the properties of their losses.


**Strengths:**

- I believe that the directions tackled by the authors is interesting, in that they propose ways to change the sampling process of projections, with the hope that it improves the properties of the discretized sliced divergences in practice.
- The contributions are thorough, with both methodological, theoretical and numerical content on their proposed sliced OT variant.

**Weaknesses:**

Major issues:

- To my mind, the use of a Markov chain does not seem to bring a significant improvement.
Concerning Table 1, after running the gradient flow, all W2 distances are of order of $10^{-5}$, whatever the choosen loss.
The running times are also equivalent up to a factor $2$.
In table 2, the FID and IS scores are not significantly different when we take into account the variance of scores (which is nice to have included it).
My take-home message of these experiments is the negative result (as an open question to prove rigorously) that there exists no sampling process which yields a significantly better error rate of the Monte-Carlo approximation.
However, I acknowledge that exploring this question of sampling is an interesting, and that even absence of significant improvement is a beneficial insight to the ML community.

- Section (3.3) on 'burning and thinning' processes is not sufficiently motivated.
First, is it a standard approach in MCMC sampling ?
If so, could the authors provide references in the main body about this ?
Second, what are the main advantages of doing this ?
There should be a key property highlighting the benefits in the main body, otherwise it looks like a variant with no main benefit.
Except the reduced computational complexity, nothing else is discussed.
If the sampling error rates scale the same as before, I am not motivated by using a more elaborate variant, and would personnally prefer to use a simple yet straightforward algorithm.
Numerical experiment also do not support a significant advantage of using these techniques.

- I have an issue with the proof of Proposition 2.
In the proof there is a term $\log (8 / \delta)$ which somehow disappears when the 'universal constant' C is defined.
However, this term seem not to converge to zero when the number of samples goes to infinity.
Could the authors provide more details on this proof ?
With all due respect to the authors, I am not completely sure of the rigor of the proof yielding this statistical rate.

Minor remarks:

- Line 51: Do we only use one projection to optimize Max-SW via gradient ascent ? It seems to be a terrible idea, probe to fall into local maxima. Also why not solving sliced Wasserstein for a set of random projections and then keep the projection yielding the largest score ?

- Line 62: Do we really need in practice to have a loss which is a distance ? In practice training a model by minimizing a loss which is only definite (e.g. Kullback-Leibler) seems to be suffice.

- Line 222: From what I understand, L is the number of Monte-Carlo chains, and T is the number of projections for a given MC chain. Thus the number of total projections is $T \times L$.

- Line 251: Maybe a notation 'Proj' and not 'Prod' would be more relevant to denote a projection operation.

- Line 263: How do you tune $\kappa$ in the Input-Aware sampling ? Do we need to scale it w.r.t. loss, data, or something else ? Some insights would be welcome here.

- In general figures have legends which are too small to read without zooming significantly. Could the authors increase their size ?

- Line 343: You could comment that this remark on having a larger T is better by connecting it to Proposition 3, since the error scales proportionnaly to $1 / T \sqrt{L}$. Theory indeed show it is more efficient to take T large rather than L.


**Questions:**

See my questions which are detailed above.

**Limitations:**

They adressed the societal impact of their work.

---

> ### Author Rebuttal · Authors · 2023-08-08
>
> We would like to show our appreciation for your valuable feedback and comments.
>
> **Q16**: To my mind, the use of a Markov chain does not seem to bring a significant improvement... all W2 distances are of order of  $10^{-5}$.
>
> **A16**: This is due to the small scale of the initial supports. The scale of distances can be easily controlled by scaling up the supports by a fixed constant, and the scale of computation time can be easily controlled by adjusting the number of supports.
>
> **Q17**: In table 2, the FID and IS scores are not significantly different when we take into account the variance of scores (which is nice to have included it).
>
> **A17**: As discussed in the global rebuttal, we believe that this is an application-specific problem. In generative modeling, reducing the FID score by a large margin becomes challenging when aiming for a high-quality generative model. For reference, the original SNGAN FID score is approximately 21, which is significantly lower than the values presented in our paper. Overall, we firmly believe that iMSW demonstrates distinct advantages in terms of gradient flow and color transfer.
>
> **Q18**: On burning thinning techniques.
>
> **A18**: The introduction of burning and thinning can be found in Chapter 10 of [1]. We kindly direct the reviewer to the global rebuttal for a comprehensive discussion regarding the benefits of burning and thinning. In brief, we propose the utilization of burning and thinning as a computational acceleration method.
>
> [1] Monte Carlo theory, methods and examples, Orwen et al.
>
> **Q19**: On the proof of sample complexity.
>
> **A19**:  We apologize for the oversight in omitting the proof. We kindly refer the reviewer to a more detailed proof provided in the rebuttal PDF file. In the course of the proof, it becomes evident that the variable $\delta$ disappears as we transition from a probability-based statement to an inequality of expectations
>
>
> **Q20 **: Line 51: Do we only use one projection to optimize Max-SW via gradient ascent ? It seems to be a terrible idea, probe to fall into local maxima. Also why not solving sliced Wasserstein for a set of random projections and then keep the projection yielding the largest score ?
>
>
> **A20**: The projected gradient ascent presently stands as the most widely employed algorithm for solving Max-SW, as highlighted in references [2], [3], and [4]. Your suggestion is indeed intriguing and aligns with the motivation behind the burning and thinning technique. With burning, we constrain ourselves to projecting directions at the final time step. Your proposed approach, involving the selection of the highest score, bears resemblance to filtering and resampling techniques in Markov chain literature. However, converting this concept into a well-defined distance in a population form poses challenges. Additionally, attaining global maxima through initial random projections can be computationally demanding, particularly in high-dimensional spaces. In summary, at the moment, formulating this technique within the context of SW literature is not trivial, and we defer its exploration to future work
>
> [2] Generalized Sliced Wasserstein distances, Kolouri et al
>
> [3] Distributional Sliced-Wasserstein and Applications to Generative Modelin, Nguyen et al
>
> [4] Shedding a PAC-Bayesian Light on Adaptive Sliced-Wasserstein Distances, Ohana et al.
>
> **Q21**: Line 62: Do we really need in practice to have a loss which is a distance? In practice training a model by minimizing a loss which is only definite (e.g. Kullback-Leibler) seems to be suffice.
>
>
> **A21**: One of the key advantages of optimal transport-based distances, in comparison to KL divergence (including its generalization f-divergence), is their capacity to handle non-overlapping support measures. Particularly, when confronted with non-overlapping support measures such as point clouds or histograms, KL divergence becomes ill-defined due to density ratio issues. Additionally, symmetry holds significance when aiming for a symmetric estimator, as is the case in learning latent representations. Furthermore, triangle inequality plays a crucial role in establishing distance properties, including aspects like two-sided sample complexity. All in all, possessing the complete set of metric properties proves advantageous both in theory and practical applications.
>
>
>
> **Q22**: Line 222: From what I understand, L is the number of Monte-Carlo chains, and T is the number of projections for a given MC chain. Thus the number of total projections is TxL
>
>
> **A22**: Yes, $T$ L is the number of Monte-Carlo chains, T is the number of projections for a given MC chain, and the total number of projections is $L\times T$.
>
> **Q23**: Line 263: On choosing $\kappa$
>
> **A23**: In the paper, we select $\kappa$ from $\{10, 50, 100\}$. Here, $\kappa$ serves as a hyperparameter, influencing the exploration of the Markov chain. A lower $\kappa$ value fosters more exploratory behavior in the Markov chain, aiding the discovery of various project directions. However, excessively small $\kappa$ equates MSW to SW. On the other hand, a higher $\kappa$ value leads to a more concentrated transition distribution around the location parameter. An excessively high $\kappa$ value causes the vMF distribution to become deterministic—akin to a Dirac delta centered at the location parameter—prompting viMSW to revert to íMSW
>
> **Q24**: On the approximation error with respect to the number of time steps $T$
>
>
> **A24**: Although augmenting $T$ could mitigate the approximation error, its reduction rate hinges on the variance of the mean projected distance along the Markov chain. This variance is influenced by both the prior and transition distributions. Currently, analyzing this variance comprehensively presents challenges. Consequently, we defer this investigation to future research works.
>
> **Q25**: Typos and presentation
>
> **A25**: We will fix all typos and adjust the revision of the paper based on your suggestion.

---

> > ### Comment · Reviewer_YkFo · 2023-08-15
> > **Response**
> >
> > I thank the authors for their rebuttal. The proof seems to the best of my understanding rigorous. I also join the opinion of Reviewer 72x2 that the performance improvement of sampling dependent projections is entirely convincing, but proposing such idea might inspire new directions of research for the community. For these reasons I decide to increase my score.

---

> > > ### Author Response · Authors · 2023-08-15
> > > **Response to Reviewer**
> > >
> > > Thank you for raising the score to 7.
> > >
> > > We really appreciate your questions and constructive feedback which helps us to improve the paper. Please feel free to ask if you still have other questions.
> > >
> > > Best regards,
> > >
> > > Authors,

---

### Official Review · Reviewer_72x2 · 2023-07-03

**Soundness:** 4 excellent
**Presentation:** 4 excellent
**Contribution:** 3 good
**Rating:** 6
**Confidence:** 4

**Summary:**

This paper introduces Markovian Sliced-Wasserstein distances, which imposes a Markovian structure on the projecting directions. The author impose different distributions on the transition kernel.

**Strengths:**

- The paper is well written and easy to follow. The idea is well-presented.
- Each important theoretical quantity is studied (metricity, sample complexity, MC approximation error, computational complexity).
- Different choices of transition kernels are proposed and studied.

**Weaknesses:**

- I think experiments are not that convincing for people to adopt this method. For instance, Figure 3 graphs and Table 2 results are not enough to show that this method is better than the other ones (see standard deviations for instance).
- I think this is the type of paper that iterates in the study of SW projections, with a standard theoretical study, good experiments, well-written, but that may not be groundbreaking.

**Questions:**

- The experiments in generative modeling are not very convincing, would you have the time to do some other experiments? I think it would benefit the paper by giving more impact to the paper.
- One of the best method for smart SW projections is Distributional SW. Why did the authors did not compare their method with it?
- I am waiting to see other reviewers questions, to see if it raises other questions on my side.

I am willing to discuss and modify my score in accordance if the authors address my points.

**Limitations:**

No need to discuss societal impact here.

---

> ### Author Rebuttal · Authors · 2023-08-08
>
> We'd like to express our gratitude for the constructive feedback. We're here to provide responses as outlined below. We are eager to participate in further discussions.
>
> **Q13**: The experiments in generative modeling are not very convincing, would you have the time to do some other experiments? I think it would benefit the paper by giving more impact to the paper.
>
> **A13**: As discussed in the global rebuttal, we believe that iMSW has clearly demonstrated its benefits in gradient flow and color transfer. The primary reason for including deep generative modeling in the paper is to showcase the versatility of the proposed distances. As mentioned in the global rebuttal, we believe we have achieved the highest performance of the SNGAN architecture. Therefore, it is challenging to significantly improve the FID score by a wide margin. We have presented three applications, and iMSW outperforms the other baseline considerably in two of them. Additionally, we have included a comparison between iMSW and DSW on point-cloud interpolation in the rebuttal PDF, which can be viewed as a more practical scenario for gradient flow. In this application, iMSW has demonstrated superior performance compared to DSW. Overall, we believe our paper presents relatively intensive experiments compared to the current literature on proposing new SW variants [1] [2].
>
> [1] Generalized sliced Wasserstein distances, Kolouri et al
>
> [2] Spherical Sliced Wasserstein, Bonet et al
>
> **Q14**: I think this is the type of paper that iterates in the study of SW projections, with a standard theoretical study, good experiments, well-written, but that may not be groundbreaking.
>
> **A14**: Sliced Wasserstein (SW) distance has various applications in practice e.g., deep generative modeling, point-cloud processing [3], texture synthesis [4], clustering [5], and so on. Therefore, any improvements to SW could make a good impact on downstream applications. Moreover, in the paper, we discuss a new aspect of SW  i.e., dependent projections which has been never investigated in the literature. Overall, we believe our paper could be one key step to push forward the applications of SW variants in general.
>
> [3]  Lidar Upsampling With Sliced Wasserstein Distance, Savkin et al,
>
> [4]  A Sliced Wasserstein Loss for Neural Texture Synthesis, Heitz et al,
>
> [5] Sliced Wasserstein Distance for Learning Gaussian Mixture Models, Kolouri et al,
>
> **Q15**: One of the best method for smart SW projections is Distributional SW. Why did the authors did not compare their method with it?
>
>
> **A15**: In the paper, our focus is on the independent structure between projections, rather than the law governing projecting directions. DSW can indeed provide a reliable law of projecting directions; however, it's important to note that projecting directions are still independently and identically distributed (i.i.d.) samples from that law. In response to your suggestion, we have conducted additional experiments to compare iMSW with DSW in gradient flow, point-cloud interpolation, and deep generative modeling on CIFAR. For a comprehensive understanding of the experimental setup and results, we kindly refer the reviewer to the global rebuttal.
> As outlined in the paper, specifically in lines 240-241, MSW can utilize the law discovered by DSW as a prior for the Markov chain. This advantageous prior enables a practical reduction of the Markov chain length, constituting a noteworthy extension. We firmly believe that a more profound exploration of the prior distribution and its corresponding transition distribution merits an independent study. Consequently, we have deferred this investigation to future research endeavors.

---

> > ### Comment · Reviewer_72x2 · 2023-08-13
> > **Answer to rebuttal**
> >
> > I would like to thank the authors for their rebuttal.
> >
> > As the experiments with DSW are included in the paper, I am increasing my score by 1.
> > Overall I think that this is a good paper, well written with a good set of experiments. I am still not 100% convinced in the adoption of the method in practice, but I think that the direction of dependent projections is worth exploring.

---

> > > ### Author Response · Authors · 2023-08-13
> > > **Response to Reviewer**
> > >
> > > Thank you for your response and for increasing the score to 6.
> > >
> > > We deeply appreciate the comment "the direction of dependent projections is worth exploring.". Overall, the main goal of MSW is to improve the performance of SW in applications. Should the reviewer have any specific concerns within practical contexts, we would be delighted to engage in a more detailed discussion about the adoption of MSW.
> > >
> > > Best regards,

---

### Official Review · Reviewer_yZut · 2023-07-03

**Soundness:** 4 excellent
**Presentation:** 4 excellent
**Contribution:** 3 good
**Rating:** 7
**Confidence:** 5

**Summary:**

This paper presents a novel variant of the sliced Wasserstein (SW) distance, addressing the challenges of redundancy and loss of metricity in other SW distance variants, such as Max-K-SW. The proposed method introduces the use of Markov chain-based projection directions selection to overcome these issues.

The theoretical contributions of the paper include proving that the new variant of SW distance is a metric, establishing its ability to induce weak convergence. Additionally, sample complexity bounds are derived, demonstrating that the proposed distance, called MSW, does not suffer from the curse of dimensionality. The authors also analyze the Monte Carlo approximation error for computing the distance.

Empirical evaluations are conducted to validate the performance of MSW compared to other SW distance variants. The results demonstrate the superiority of MSW in various tasks, showcasing its effectiveness and advantages in practical applications.

**Strengths:**

- The paper presents a novel and intriguing idea of utilizing a Markov chain to select projection directions. This approach offers a fresh perspective on addressing the challenges of redundancy and loss of metricity in other variants of the sliced Wasserstein (SW) distance. The novelty and innovation of this idea contribute to the strength of the paper.
- The authors showcase a commendable strength in their thorough theoretical analysis, covering multiple aspects such as topology and statistics. This comprehensive analysis ensures that the proposed method is well-founded and supported by solid theoretical underpinnings.
- The experimental findings presented in the paper are convincing and demonstrate the superior performance of the newly introduced method compared to other variants of the SW distance. However, I have some questions regarding these experiments and see below.

**Weaknesses:**

One potential weakness of the paper lies in the claim that Max-K-SW suffers from a loss of metricity when not obtaining the optimal projections. While this issue is acknowledged, it would be beneficial for the authors to emphasize how their proposed method, even with the use of Monte Carlo approximation, mitigates or eliminates this problem.

**Questions:**

- It would be valuable if the authors could provide further explanation regarding why the newly introduced method outperforms other variants of the sliced Wasserstein (SW) distance empirically. Considering Proposition 1, which suggests that Max-SW is a better proxy for the Wasserstein distance, is it due to the errors introduced in computing the optimal projection direction? Additional insights into the specific factors that contribute to the superior performance would enhance the understanding of the proposed method's advantages.

typos
- line 593: "We samples" should be "We sample"
- line 703: in the limit I assume that you mean $\mu_{\phi(k)}$ instead of $\mu_k$.
- line 704: in the limit, $(\mu_k,\mu)$ shouldn't be in subscript. In general the subsequence notation is weird in this proof. For example, $\phi$ appears twice in the proof with different meanings. Please fix this.
- line 727: do you mean $\Theta$ is the compact support of $\mu$?
- line 735: it's better to write something like $\mathcal{B}_\theta$ to make the set depending on $\theta$ explicitly.

---

> ### Author Rebuttal · Authors · 2023-08-08
>
> We begin by expressing our gratitude for the insightful feedback and comments provided in the review. In response, we offer the following explanations and answers to your inquiries. We welcome any further questions and are open to engaging in a more in-depth discussion on the matter.
>
> **Q9**: One potential weakness of the paper lies in the claim that Max-K-SW suffers from a loss of metricity when not obtaining the optimal projections. While this issue is acknowledged, it would be beneficial for the authors to emphasize how their proposed method, even with the use of Monte Carlo approximation, mitigates or eliminates this problem.
>
>
> **A9**: Thank you for your comment. The difference becomes apparent when considering the asymptotic setting. In particular, for Max-K-SW, it cannot guarantee metricity even as $T$ approaches $\infty$ due to the non-convex optimization problem. In contrast, as $L$ approaches $\infty$, MSW can guarantee metricity due to the law of large numbers. Moreover, Max-K-SW is deterministic in nature; hence, there is no derived approximation guarantee. However, we can determine the approximation error of MSW thanks to the MC approximation. Additionally, the unbiasedness of the MC approximation could lead to a better optimization loss, a phenomenon widely observed in SGD.
>
>
> **Q10**: It would be valuable if the authors could provide further explanation regarding why the newly introduced method outperforms other variants of the sliced Wasserstein (SW) distance empirically. Considering Proposition 1, which suggests that Max-SW is a better proxy for the Wasserstein distance, is it due to the errors introduced in computing the optimal projection direction? Additional insights into the specific factors that contribute to the superior performance would enhance the understanding of the proposed method's advantages.
>
>
> **A10**: Proposition 1 implies that MSW can retain the favorable properties of SW and Max-SW, such as low sample complexity. In the paper, we do not contest the optimality of Max-SW in the population form. However, the practical limitations of the current optimization scheme prevent Max-SW from achieving its optimum. Furthermore, utilizing Max-SW as a minimization loss gives rise to a challenging min-max problem, well-known for its difficulty. Motivated by these challenges, we introduce MSW as a computationally effective and efficient variant that establishes a flexible framework for selecting projecting directions. The MSW population offers expressiveness in terms of prior distribution selection, transition distribution, and the lengths of the Markov chain ($T$). Therefore, we could balance between exploration in the space of projecting directions and searching for good projecting directions as we do with iMSW.
>
> On the theoretical side, there exist numerous theoretical challenges in comprehending MSW due to its expressiveness, hence, we believe it is worth careful independent future exploration. In this paper, we concentrate on the fundamental properties of MSW, demonstrating its practicality and its applicability in various applications. Therefore, we suggest viewing MSW as a practical variant of SW at the moment.
>
>
> **Q11**:  do you mean $\Theta$ is the compact support of $\mu$?
>
>
> **A11**: Yes, we will revise the proof to indicate it explicitly.
>
> **Q12**: On typos and presentation suggestions.
>
>
> **A12**: Thank you for pointing this out. We will fix all the typos and adjust our presentation based on your suggestion in the revision.

---

> > ### Comment · Reviewer_yZut · 2023-08-16
> >
> > Thank you for addressing my questions.
> >
> > I would like to clarify a little of my question. I was referring to Proposition 1 (i) in the manuscript which is talking about some inequalities related different metrics instead of sampling complexity mentioned in proposition 2. \
> >
> > From this proposition, it seems that Max-SW is theoretically closer to the Wasserstein distance, suggesting it might be a more accurate theoretical proxy. While I acknowledge the authors' arguments regarding the computational challenges of Max-SW, I'm curious if there are any practical experiments that demonstrate that the inequality MSW<Max-SW does not hold consistently in practice. Showcasing such practical limitations would further substantiate the superiority of the proposed method over Max-SW.

---

> > > ### Author Response · Authors · 2023-08-16
> > > **Response to Reviewer**
> > >
> > > Thank you for a very interesting suggestion. This is the type of experiment that we did not have in our paper.
> > >
> > > We have calculated the values of iMSW and Max-SW between two point clouds as in the rebuttal setting. We use the same setting as in the rebuttal, and we get the maximal value of Max-SW is 12.7 and the maximal value of iMSW is 23.2. Therefore, it seems that iMSW reaches a "better" projecting direction than the Max-SW by using two Markov chains. Nevertheless, despite the fact that Max-SW seems to be a better proxy for Wasserstein distance in theory, Max-SW might not be robust in the setting of noisy and contaminated distribution, which might happen in large datasets e.g., CIFAR10, CelebA. In this setup, the finding of the max projecting direction could be affected significantly by the noise.  Overall,  we will add this report and the discussion to our revision.
> > >
> > > We are happy to discuss this further if the reviewer is still not satisfied with the answer. Also, please feel free to ask if the reviewer still has questions.
> > >
> > > Best regards,

---

> > > > ### Comment · Reviewer_yZut · 2023-08-19
> > > >
> > > > I'm satisfied with the answer and I would like to maintain my supportive score for the paper.

---

> > > > > ### Author Response · Authors · 2023-08-19
> > > > > **Response to Reviewer**
> > > > >
> > > > > We would like to thank the reviewer for the supportive score. We will include all the discussion in the revision.
> > > > >
> > > > > Best regards,
> > > > >
> > > > > Authors,

---

### Official Review · Reviewer_AyBm · 2023-07-04

**Soundness:** 3 good
**Presentation:** 3 good
**Contribution:** 3 good
**Rating:** 6
**Confidence:** 3

**Summary:**

The authors propose a variant of the sliced Wasserstein distance that samples the projection directions from a Markov chain instead of uniformly from the hypersphere. They propose two Markov chain designs: the first chooses the next projection vector to be orthogonal to the previous one and the second is deterministic and basically performs projected gradient ascent on the projected Wasserstein distance.

**Strengths:**

- The authors analyze their proposed method theoretically in various ways
- Multiple experiments compare the proposed method against baselines

**Weaknesses:**

1. In high dimensions, two random vectors will be almost orthogonal to each other with high probability. Therefore, oMSW should be basically equivalent to SW which is confirmed by Table 2 and Table 4 in the appendix (compare SW(L=45) to oMSW(L=3,T=15), so that they use the same number of samples; both methods take roughly the same time and lead to almost the same Wasserstein-2 distance)
1. Table 2: The majority of results overlap so much (i.e. many are within one standard deviation of each other) that no conclusions can be drawn. In particular, putting any of the numbers in bold is misleading.
1. oMSW is only evaluated in Table 2 (which does not really show anything, see previous point)
1. Burned-thinned MSW is introduced as a performance enhancement but (1) its impact only evaluated in the appendix (2) its impact on performance is difficult to understand from Table 3 and it looks like there is no real benefit to it
1. Paragraph "Stationarity of $\sigma_T(\theta_T)" is has neither positive nor negative results and is vague overall

**Questions:**

1. How do you choose the parameters of the vMF distribution? Why would there be any preference regarding the first projection direction?
1. Why would the stationary distribution of the $\sigma_T(\theta_T)$ for uniform initialization and orthogonal chain not be the uniform distribution?
1. If orthogonality of directions is a desirable property, shouldn't oMSW always be worse than K-SW for $T = K$?

**Limitations:**

The authors did not discuss any limitations of their method.

---

> ### Author Rebuttal · Authors · 2023-08-08
>
> First, we would like to thank the reviewer for the time and constructive feedback.
>
> **Q1**: In high dimensions, two random vectors will be almost orthogonal with high probability. Therefore, oMSW should be basically equivalent to SW.
>
> **A1**: Regarding the concentration result, two uniformly drawn vectors from the unit-hypersphere are likely to be orthogonal in high dimensions. However, when $2 << K \approx d$, uniformly drawn vectors are not likely to be orthogonal with high probability. Moreover, there are also applications on low dimensions e.g., 3D point-clouds. In Table 4, oMSW (L=3, T=5) has a W2 score of 415.06, while SW (L=15) has a score of 421.5. Therefore, they are not the same for a limited number of projections. Since the key motivation of MSW is to reduce the number of projections in practice, we believe orthogonal sampling could still offer some benefits. We acknowledge that the performance of the orthogonal transition is not as appealing as the input-aware transition (iMSW). However, we believe that discussing the orthogonal transition is interesting and helpful to future works.
>
>
> **Q2**: Table 2: The majority of results overlap so much.
>
> **A2**: We believe this is an application-specific problem. In generative modeling, achieving a significant reduction in the FID score when reaching a high-quality generative model is challenging. As a reference, the original SNGAN had a FID score of about 21, which is not as favorable as the results presented in our paper. We offer generative modeling mainly as evidence of the widely applicable capabilities of the proposed distances. Additionally, in the paper, we highlight that iMSW demonstrates superior performance in gradient flow and color transfer applications.
>
> **Q3**: oMSW is only evaluated in Table 2
>
> **A3**:   In the gradient flow application, $d=2$, hence oMSW is equivalent to K-SW.This equivalence is mentioned in line 338 of the submission. As previously discussed in **Q1*, we believe that oMSW offers advantages in practical scenarios involving a restricted number of projections. Furthermore, it's important to note that oMSW is just one variant of MSW. In the paper, we also introduce iMSW, which achieves significantly better results compared to oMSW.
>
> **Q4**: On Burned-thinned MSW
>
> **A4**: In summary, from Table 1, iMSW (L=2, T=5) demonstrates a W2 value of 0.0064 and a runtime of 1.41 seconds. Moving to Table 3, iMSW (L=2, T=5, M=4) achieves a W2 of 0.0101 and takes 1.2 seconds; iMSW (L=2, T=5, M=0, N=2) shows a W2 of 0.0066 and a runtime of 1.28 seconds; iMSW (L=2, T=5, M=2, N=2) yields a W2 of 0.0072 and completes in 1.19 seconds. These results collectively indicate that employing burning and thinning aids in reducing computational time, a phenomenon also observed in Table 4. Furthermore, under specific configurations of burning and thinning, improved results are evident, such as iMSW (L=2, T=5, M=2) with a W2 of 0.0055 and a runtime of 1.25 seconds in gradient flow and deep generative modeling (as demonstrated in Table 5). However, burning and thinning are primarily recommended as computational reduction techniques.
>
>
> **Q5**: On paragraph "Stationarity of $\sigma_T(\theta_T)$"
>
>
> **A5**: Our main goal is to discuss the distinctions between iMSW and Max-SW, while also exploring the relationship between the two. Through this discussion, we could shed light on why iMSW outperforms Max-SW, partially attributed to the stationarity of iMSW—characterized as a mixture of local minima from the optimization problems within Max-SW.
>
>
> **Q6**: How do you choose the parameters of the vMF distribution?
>
>
> **A6**: For gradient flow and color transfer applications, we choose values of $\kappa$ from the set $\{10, 50, 100\}$. In the context of generative modeling, our selection for $\kappa$ comes from ${10, 50}$. Conceptually, the concentration parameter $\kappa$ in viMSW can be interpreted as an exploration parameter. Specifically, when $\kappa$ is small, the transition distribution explores the vicinity of the location parameter more extensively. As $\kappa$ approaches 0, the transition distribution tends towards a uniform distribution over the unit-hypersphere. On the other hand, higher values of $\kappa$ result in a more adhesive behavior of the transition distribution around the location parameter. When $\kappa$ tends towards infinity, the transition distribution becomes deterministic, leading viMSW to converge back to iMSW.
>
>
> **Q7**: Why would there be any preference regarding the first projection direction?
>
> A7:  From a theoretical standpoint, employing a continuous prior distribution ensures the metricity of the distance, applicable to both deterministic and probabilistic transitions. Looking from an empirical perspective, a well-chosen prior distribution can effectively determine the length of the chain, denoted as $T$, allowing for shorter chain lengths due to the availability of a favorable distribution governing projecting directions.
>
>
> **Q8**: Why would the stationary distribution of the $\sigma_T(\theta_T)$ for uniform initialization and orthogonal chain not be the uniform distribution?
>
> **A8**: Thank you for your insightful question. Addressing this fundamental question isn't straightforward, and at present, we lack a definitive answer. However, as elaborated upon in **Q7**, it's feasible to employ alternative non-uniform distributions as the prior distribution. The choice of a uniform prior in the paper primarily stems from its simplicity and continuity.
>
>
> **Q9**: If orthogonality of directions is a desirable property, shouldn't oMSW always be worse than K-SW for $T=K$
>
> **A9**: If prioritizing the orthogonality of directions is a key objective, K-SW would outperform oMSW when considering the same number of projections. However, it's important to note that oMSW offers a computational advantage over K-SW, as enforcing pairwise orthogonality in K-SW comes with a quadratic increase in computational complexity.

---

> > ### Comment · Reviewer_AyBm · 2023-08-13
> >
> > Thank you for the clarifications. I have raised my score accordingly.

---

> > > ### Author Response · Authors · 2023-08-13
> > > **Response to Reviewer**
> > >
> > > Thank you for raising the score to 6.
> > >
> > > Please feel free to ask if the reviewer still has questions. We are happy to make further discussion.
> > >
> > > Best regards,

---

### Author Rebuttal · Authors · 2023-08-08

We would like to begin by extending our gratitude to the reviewers for their valuable time and feedback. In our response, we will outline a summary of additional experiments in the rebuttal PDF and provide answers to questions frequently raised by the reviewers. Further inquiries have been addressed in the corresponding rebuttal aimed at addressing the concerns of the reviewers.

1. **Detailed proof for sample complexity.** In response to the suggestion of Reviewer **YkFo**, we have provided a more detailed derivation of the sample complexity proof in the rebuttal PDF. Specifically, we discuss how we can derive the interested expectation inequality from the original probability statement of Vapnik–Chervonenkis theorem using the Jensen inequality, tail sum expectation, and change of variables.

2. **Comparison to DSW**. As suggested by Reviewer **72x2**, we have added a comparison to DSW [1]. DSW is a notable variant of SW that selects the most discriminative slicing distribution in a family of distributions over the unit-hypersphere. However, DSW still utilizes independent and identically distributed (i.i.d) projecting directions. In contrast, our focus is on constructing dependent projecting directions via the first-order Markov structure. We provide new experiments on gradient flow to compare iMSW and DSW. We set $L=2, T=5$ for both iMSW and DSW. For iMSW, we use $\eta = 0.1$. For DSW, we report the best hyperparameters: $\lambda \in {1,10,50}$ and slice learning rate $\eta \in {0.001,0.01,0.1}$. Figure 1 in the rebuttal PDF shows that iMSW still drives the flow to converge faster than DSW. For a more realistic usage of gradient flow, we repeated the comparison using point-cloud interpolation with two randomly selected point-clouds from ShapeNet [2], each containing 2048 points. Overall, we still observe the same phenomenon: iMSW is faster and performs better. Finally, we present the results of DSW in deep generative modeling on CIFAR10 with FID 14.18 and IS 8.14, which are slightly worse than iMSW. Due to time constraints, we were unable to run the experiments multiple times and conduct experiments on CelebA.

    As discussed in the paper, DSW can be used as a building block for MSW. For instance, the slicing distribution found in DSW could serve as a good prior distribution. Moreover, we can also design a transition distribution in a similar way as DSW. We would like to emphasize that MSW is the first variant of SW that discusses non-stationary slicing distribution, offering a flexible modeling opportunity. Given the extensive room for designing better prior and transition distributions based on recent literature [1], [3], we leave a detailed investigation for future work.

   [1] Distributional Sliced-Wasserstein and Applications to Generative Modeling, Nguyen et al

   [2] https://shapenet.org/

    [3] Shedding a PAC-Bayesian Light on Adaptive Sliced-Wasserstein Distances, Ohana et al.

3. **On the significance of experimental results.** As discussed with Reviewers **YkFo** and **72x2**, we would like to emphasize the significance of the experimental results. We believe iMSW has demonstrated significantly better performance in gradient flow and color transfer. Particularly, iMSW reduces the Wasserstein metric by about 91% compared to SW and by at least 50% compared to Max-SW and Max-K-SW at step 200 in gradient flow (Figure 1 and Figure 2), all while being faster. In color transfer (Figure 2), iMSW reduces Wasserstein-2 by at least 95% compared to SW and Max-(K)-SW. Additionally, it is worth noting that the scale of the distances can be easily controlled by scaling the supports of measures by a constant. In generative modeling, significantly reducing the FID score becomes challenging when reaching a high-quality generative model. For reference, the original SNGAN FID score is about 21 which is considerably worse than those reported in our paper. Overall, we provide these experiments mainly as evidence of the widely applicable capability of the proposed distances.

4. **On burning and thinning.** As discussed with Reviewers **AyBm** and **YkFo**, burning and thinning techniques could reduce computational time while maintaining comparable performance. For instance, in gradient flow, iMSW (L=2, T=5) achieves W2=0.0064 and takes 1.41 seconds, iMSW (L=2, T=5, M=0, N=2) achieves W2=0.0066 and takes 1.28 seconds, and iMSW (L=2, T=5, M=2, N=2) achieves W2=0.0072 and takes 1.19 seconds. A similar phenomenon is observed in color transfer (Table 4); for example, iMSW (L=3, T=15) achieves W2=15.23 and takes 79.47 seconds, iMSW (L=3, T=15, M=0, N=2) achieves W2=16.6 and takes 62.66 seconds, and so on. Interestingly, burning and thinning can improve performance with the right choice of hyperparameters (Table 1, Table 3, and Table 5). However, we still recommend using burning and thinning as computational acceleration techniques due to their widely recognized benefits.

Overall, we would like to reiterate that the main focus of the paper is to provide a unified approach to using dependent projecting directions for the sliced Wasserstein distance. Specifically, we offer a clear population formulation that has not been previously derived, along with a discussion of its theoretical properties. While the paper presents specific instance choices of the prior distribution and transition distribution, yielding notable performance (iMSW), the designs proposed may not be optimal and could be further improved. Given the flexibility of MSW in designing the Markov structure, we leave a thorough investigation to future work.

Best regards,

Authors,

---

### Comment · Area_Chair_gcWm · 2023-08-11
**Discussions**

 Dear reviewers and authors,

Thank you very much for your work on this submission and its evaluation. Now that the authors have responded to the reviews, I **strongly encourage** the reviewers to acknowledge the review, to look at other reviews and rebuttals for this submission, and to adjust their scores if needed. Thanks to those that have already done so.

Authors have the possibility to reply if further questions are needed, until the 16th.

Thank you very much to all,

Area Chair

---

### Decision · Program_Chairs · 2023-09-21

**Decision:**

Accept (poster)

**Comment:**

The paper proposes a novel way for doing sliced Wasserstein. Instead of using an iid sampling from a
predefined distribution on the sphere, they propose to use a first-order Markovian structure on random
84 projecting directions. Several properties of the new method have been proved and its empirical performance
compared to other sliced methods has been highlighted.
Overall, all reviewers are happy about the paper and the rebuttal and propose to accept the paper.